# Simbi: historical hydro-meteorological time series and signatures for 24 catchments in Haiti

Ralph Bathelemy[1,2*], Pierre Brigode[1,3,4], Vazken Andréassian[3], Charles Perrin[3], Vincent Moron[5], Cédric Gaucherel[6], Emmanuel Tric[1], Dominique Boisson[2]

[1] Université Côte d'Azur, Observatoire de la Côte d'Azur, CNRS, IRD, Géoazur, France

[2] Université d'État d'Haïti, Faculté des Sciences, LMI CARIBACT, Urgéo, Haïti

[3] Université Paris-Saclay, INRAE, HYCAR, Antony, France

[4] Univ Rennes, CNRS, Géosciences Rennes - UMR 6118, F-35000 Rennes, France

[5] Aix Marseille University, CNRS, IRD, INRAE, CEREGE, Aix-en-Provence, France

[6] AMAP, INRAE, University of Montpellier, CNRS, IRD, Cirad, Montpellier, France

*correspondence: Ralph Bathelemy (bathelemyr@gmail.com)

## Abstract

Haiti, a Caribbean country, is highly vulnerable to hydroclimatic hazards due to heavy rainfall, which is partly linked to tropical cyclones. Additionally, its steep slopes generate flash floods, particularly in small catchments. Moreover, the hydrology of this region remains poorly understood and understudied. Unfortunately, there is no accessible database for the scientific community to use in this country. To fill this gap, hydroclimatic data were collected to create the first historical database in Haiti. This database, called "Simbi" (guardian of rivers, freshwater, and rain in Haitian mythology), includes 156 monthly rainfall series over the period 1905–2005, 59 daily rainfall series over the period 1920–1940, 70 daily streamflow series, and 23 monthly temperature series, not necessarily continuous, over the period 1920–1940. It also provides simulated streamflow series over the period 1920–1940 using the GR2M and GR4J rainfall–runoff models for 24 catchments and 49 attributes covering a wide range of topographic, climatic, geological, land use, hydrogeological, and hydrological signature indices. Simbi is the first open-access hydro-meteorological dataset for Haiti and will contribute to a better knowledge of hydrological risk in Haiti. Several sources of uncertainty associated with Simbi are acknowledged, including data quality (historical data), digitisation of paper archives, identification of relevant raingauges, and rainfall-runoff models. It is important to consider these uncertainties when using Simbi.

The database will be regularly updated to include additional historical data that will be digitized in the future. It will thus contribute toward better knowledge of the hydrology of Haitian catchments and will enable the implementation of various hydrological calculations useful for designing structures or flow forecasting. Simbi is an open access database and is available for download at: https://doi.org/10.23708/02POK6 (Bathelemy et al., 2023).

## 1   Introduction

Hydroclimatic databases, generally composed of climatic (precipitation and air temperature) and hydrological (streamflow) time series at the catchment scale, are extremely useful (Tramblay et al., 2021). They are used for water resources planning and management as well as for monitoring and forecasting floods, droughts, and changes in surface and groundwater resources (Dewandel et al., 2003, 2004; Alfieri et al., 2020; Harrigan et al., 2020).These databases are also used to evaluate the performance of "new" hydro-meteorological products based on Earth observation satellites, which are increasingly applied in poorly instrumented regions (Beck et al., 2019; Brocca et al., 2019; Prakash, 2019; Bathelemy et al., 2022). Furthermore, they are central to studies of climate change impact, for example, through the calibration and evaluation of hydrological models used to quantify climate change impacts on water resources (Abbaspour et al., 2009; Chokkavarapu and Mandla, 2019; Teutschbein and Seibert, 2012).

In recent years, hydroclimatic databases called "CAMELS" (catchment attributes and meteorology for large-sample studies) have been created in several countries: United States (Addor et al., 2017), Chile (Alvarez-Garreton et al., 2018), Brazil (Chagas et al., 2020), Great Britain (Coxon et al., 2020), Australia (Fowler *et al.*, 2021), Central Europe (Klingler et al., 2021), and Switzerland (Höge et al., 2023). The CAMELS databases
use large datasets (precipitation, streamflow, air temperature, etc.) from multiple sources (in situ, reanalysis, remote sensing, etc.) over several hundreds of catchments. They also include multiple catchment attributes covering a range of topographic, climatic, hydrological, geological, and land cover indices. While the CAMELS databases provide time series, indices and hydroclimatic signatures of catchments, other databases provide only indices and hydroclimatic signatures of catchments, such as the African Database of Hydrometric Indices
(ADHI; Tramblay *et al.*, 2021). These databases give the scientific community easy access to the hydrological information available for the regions concerned.

Unfortunately, there are significant differences between countries in terms of the quality and quantity of hydroclimatic reference databases, as well as regarding access to these data. Some countries do not have such reference databases. This is the case of Haiti, whose territory is, moreover, highly exposed to natural
disasters (Khouakhi et al., 2017; Burgess et al., 2018), and climate change (Peterson et al., 2002). At the same time, Haiti is facing the consequences of massive deforestation and anarchic urbanization (urban development that does not comply with planning regulations) in recent decades (Hedges et al., 2018; Tarter et al., 2018; Mompremier et al., 2022), resulting in increased vulnerability to hydroclimatic hazards. Currently, Haiti lacks a freely and easily accessible hydroclimatic database due to the absence of in situ hydroclimatic
observations. The first hydrometric observations were conducted during the American occupation of Haiti, and began in 1919. American engineers from the Water Resources Service (WRS) of the United States Geological Survey (USGS) supervised these hydrological observations, that continued into the 1940s and exceptionally later. The end of the American occupation is the main reason for the cessation of hydrometric observations. This is due to the loss of technical support from the WRS, as well as financial constraints and socio-political
difficulties in Haiti. The data time series and a description of the methods used to collect them were published annually in the "Hydrographic Bulletin", summarizing 70 daily streamflow time series over the 1920-1940 period. After these two decades of streamflow observations, very few hydrological data were produced in Haiti (Pouyaud and Hoepffner, 1987). In addition to hydrometric observations, rainfall measurements started in Haiti around 1905 using 15 raingauges. Over time, the raingauge network became denser, with 25 stations
operated by the "Petit-Séminaire Collège St Martial" (a school run by the Congrégation du Saint-Esprit), 38 by the "Direction Générale des Travaux Publics", and nearly 30 by other institutions, such as the" Frères de l'Instruction Chrétienne" (Pouyaud and Hoepffner, 1987). Rainfall measurements are currently managed by the CNIGS and the UHM. Since 2014, this observation network has had approximately twenty automatic raingauges. However, due to a significant amount of missing data, the network remains highly fragmentary
and unexploited.

In 1977, the Haitian government initiated a project to make an inventory and digitize some available hydroclimatic time series. As a result, the 70 daily streamflow series for the period 1920-1940 and almost a hundred monthly rainfall series from the start of observations (~1905) until 1975 were digitized. In 2012, the Haitian government launched a second project named BVH (*Bassins Versants Haïtien* in French, i.e., Haitian
catchments; Gaucherel *et al.* 2018) for compiling available hydroclimatic data, better understanding hydrology in Haiti and improving the management of water resources. Within this project, Haitian catchments were characterized using monthly streamflow data (Gaucherel *et al.*, 2016) and rainfall data (Moron et al., 2015) and the relationships between their shape, relief, and river sinuosity were investigated (Gaucherel *et al.*, 2017, Bonhomme *et al.*, 2013).Unfortunately, the two databases produced within the BVH project (monthly rainfall
time series and monthly streamflow time series) have never been analyzed jointly, are not available online and remain limited for several hydrological analysis due to their monthly time step (monthly). Thus, these databases are underused to date.

The main objectives of this study are to make Haitian hydroclimatic data available to the scientific community and to merge these different datasets in order to propose the first hydroclimatic database for several Haitian
catchments at both monthly and daily timesteps. To overcome the issue of the numerous missing data present within the streamflow time series, two rainfall-runoff models were used to reconstruct the missing values and

produce continuous streamflow time series. (Brigode et al., 2016; Smith et al., 2019). The use of rainfall-models for flow reconstruction has been used for several decades for various types of catchments covering different climatic regions (Caillouet et al., 2017; Crooks and Kay, 2015; Jones and Lister, 1998). Thus, a monthly (GR2M, Mouelhi *et al.*, 2006) and a daily (GR4J, Perrin *et al.*, 2003) lumped rainfall–runoff model were used to reconstruct continuous streamflow series in Haiti at both time steps.

The goal of our study is therefore fourfold:

    i)        Collecting all existing hydroclimatic time series in Haiti and digitizing certain paper archives that have been identified as priorities,

    ii)       building climatic (air temperature and rainfall) time series at the catchment scale by spatially and temporally aggregating available series,

    iii)     creating a continuous Haitian hydroclimatic database for the 1920–1940 period, using the catchment climatic series and the rainfall-runoff models,

    iv)     characterizing the hydrological behavior of Haitian catchments based on 49 hydrological indices and signatures covering six classes of catchment attributes (topographic, geological, hydrogeological, land cover, climate indices, and hydrological signatures).

Observed hydroclimatic data, simulated streamflow series, and catchment attributes make up the Simbi database, the first continuous and freely available hydrological database in Haiti. Simbi is a guardian of rivers, freshwater and rain in Haitian mythology (https://en.wikipedia.org/wiki/Simbi).

## 2    Data used

### 2.1    Streamflow

The streamflow data consist of 70 daily series, most of which are available from 1920 to 1940, with significant gaps (missing data) in some series (see Figure 1). These data were collected by the *Hydrographic Department of the Irrigation Service of the General Direction of Public Works* in Haiti. On average, 12 gauging measurements were performed per station and per year. These data digitized in 1977.

### 2.2    Rainfall

#### *2.2.1    Monthly rainfall*

A modified version of the monthly rainfall database compiled by Moron *et al.* (2015) has been produced for this study. The original data produced by Moron *et al.* (2015) was compared with the digitized data, allowing for correction of some data series. The modifications to the original database are described in detail in Appendix A. The original data produced by Moron *et al.* (2015) included 156 monthly rainfall series available from 1905 to 2005 and were derived from three different sources:

1.    The CNIGS (*National Center for Geospatial Information*) database with 162 monthly rainfall series;

2.    The database managed by the international company *Chemonics* with 109 monthly rainfall series;

3.    The CNSA (*National Coordination for Food Security*) database with 14 monthly rainfall series.

These three databases were merged by Moron *et al.* (2015) by removing and/or correcting duplicates. In total, 156 monthly rainfall series were validated and retained to form the monthly rainfall database. However, several series have a high percentage of missing data, and most of them have data available only from 1930 to 1970 (see Figure 1).

### 2.2.2 Daily rainfall

Nearly 15 paper registers containing hydroclimatic data between 1905 and 1970 have been recovered from the BHS (*Bibliothèque Haïtienne des Spiritains in French i.e Haitian* spiritual library). These data were previously collected by the observatory of the Petit-Séminaire Collège St-Martial (PSCSM) in Port-au-Prince during the 20th century. The daily rainfall time series considered as priority in this study are those available for the studied catchments (i.e. raingauges located within or close to the studied catchments) and for the same period as the streamflow time series, i.e. the period 1920-1940. Overall, 59 rainfall times series available for the period 1920–1940 have been digitized for the Simbi database (see Figure 1). Various optical character recognition tools were tested to perform this digitization. However, the results were not satisfactory due to the poor readability of the documents, which were both secular and handwritten. Therefore, the daily data was transcribed manually. Four students from UEH (Université d'Etat d'Haïti in French i.e State University of Haiti) were recruited for this digitization task. They worked in pairs, with one student reading and the other entering the data.

Note that the monthly database created by Moron *et al.* (2015) resulted from an initial digitization of monthly totals from the same raingauges. Thus, our digitization work extends the efforts of Moron *et al.* (2015) to assess the quality of the digitized daily rainfall series (referred as BHS hereafter), we compared their monthly sums with the Moron *et al.* (2015) database (referred as MORON hereafter). We used two criteria to compare the BHS with the MORON data:

1.   the correlation between the BHS and MORON monthly series,

2.   the percentage of months where the errors between the monthly BHS and MORON data are greater than 5%.

The BHS data was reviewed and corrected for each month in which discrepancies were found between the BHS and MORON data. It is worth noting that some errors were identified in the MORON data. The Appendix A summarizes the 5 types of errors detected in the Moron *et al.*, 2015, such as the correction of some extreme values (e.g. the precipitation of October 1933 at Camp-Perrin was equal to 196.9 mm in Moron *et al.* (2015), while the digitization of the daily data confirmed that the actual monthly sum was 1196.9 mm!). Therefore, this study allowed for the correction of both the BHS and MORON data (see Appendix A).

### 2.2.3 NOAA 20CR reanalysis rainfall

The third version of the *National Oceanic and Atmospheric Administration* (NOAA) Twentieth Century Reanalysis (20CR) project precipitation data (Slivinski et al., 2019) was used for the period 1920–1940. These data are available at a daily time step at a spatial scale of 1° (111 km at the equator). These are not measured data, but rainfall data from a global climatic model (reanalysis).

## 2.3   Air temperature

### 2.3.1 Digitization of historical archives

Air temperature data are available at a monthly time step in paper archives in the same river bulletins that contain streamflow data. A total of 23 monthly temperature series with data available for the period 1926–1939 have been digitized by three students from UCA (Université Côte d'Azur) for the Simbi database. These temperature series are not continuous over time and there are significant gaps (missing data) in some series (see Figure 1).

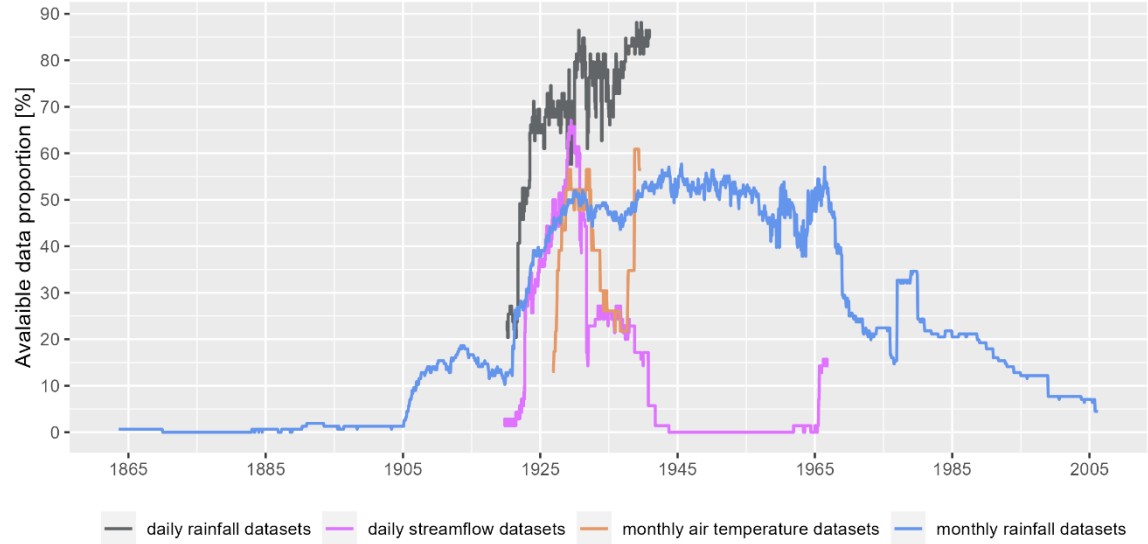

Figure 1 – Period of availability and percentage of stations with data available for digitized daily rainfall datasets, daily streamflow datasets, monthly air temperature datasets, and monthly rainfall datasets produced by Moron et al., (2015).

### 2.3.2 NOAA 20CR reanalysis air temperature

The NOAA reanalysis air temperature database (Slivinski et al., 2019) was used in this study. This air temperature database is available at the same spatiotemporal resolution as the NOAA rainfall data (see section 2.2.3).

### 2.3.3 Berkeley Earth Surface Temperature (BEST)

The BEST (Rohde et al., 2013) air temperature database was used in this study. BEST is a gridded air temperature produced by spatial interpolation using the kriging method (Krige, 1951; Cressie, 1990) of air temperature data observed around the world. BEST is starting in 1753 at the monthly resolution, and in 1880 at the daily resolution, with a 1° spatial resolution.

## 2.4 Digital Elevation Model (DEM)

The digital elevation model used in this study is the Shuttle Radar Topography Mission (SRTM) of the *United States Geological Survey* (USGS) and the *National Aeronautics and Space Administration* (NASA). The digital elevation model was extracted for Haiti and is available at a spatial resolution of 90 m (Reuter et al., 2007).

## 3 Methodology

This section presents the methodology followed (i) to select the hydrometric stations and climatic series used 195 to produce the time series at catchment scale, (ii) to simulate continuous streamflow series with rainfall-runoff models for the selected catchments, and (iii) to compute hydrological indices and signatures for the selected catchments.

The conceptual lumped GR2M and GR4J rainfall-runoff models are described in Appendix B. The KGE objective function (Kling & Gupta efficiency; Gupta *et al.* 2009) was used to evaluate the performance of both 200 models. The KGE score is defined by the following analytical formula:

$$KGE = 1 - \sqrt{(1-r)^2 + (1-\alpha)^2 + (1-\beta)^2} \qquad (1)$$

where $r$ is the correlation coefficient, $\alpha$ is the ratio of the standard deviation of the simulated streamflow to the standard deviation of the observed streamflow, and $\beta$ is the ratio of the mean of the simulated streamflow to the mean of the observed streamflow.

## 3.1    Selection of streamflow data and catchments

### 3.1.1    Selection of streamflow series

An analysis of the 70 available streamflow series was performed to select the "hydrologically relevant" streamflow series. Four criteria were initially used to make this selection:

1.  The annual hydrographic bulletins reported the accuracy with which rating curves were established through three ratings: "well established," "fairly well established," and "poorly established." Most of the streamflow series with "poorly established" rating curves were found to have significant measurement differences between periods. These streamflow series were not used in the remainder of this study.

2.  Some hydrometric stations were located downstream of diversion channels or small dams used for irrigation. These streamflow series poorly represent the seasonality of streamflow, and are therefore considered to be influenced by human activities. These streamflow series were not used in the remainder of this study.

3.  Some hydrometric stations were located downstream of resurgences or springs. These groundwater resurgences are beyond the scope of this study. Therefore, these streamflow series were not used in the remainder of this study.

4.  The streamflow series that had less than 5 years of data were not used in the remainder of this study.

In addition to these four criteria, three other indices inspired by the paper of Gudmundsson et al. (2018) were used to assess the quality of the streamflow data. These three criteria were calculated as follow:

1.  Number of days for which Q<0, where Q denotes a daily streamflow value. The rationale underlying this rule is that streamflow values smaller than zero are non-physical (Gudmundsson and Seneviratne, 2016).

2.  Sequence of more than 10 equal consecutive streamflow values larger than zero. This index was selected because equal consecutive streamflow values often occur due to instrument failure or flow regulation (Gudmundsson et al., 2018).

3.  Detection of outliers, i.e. unusually large or small streamflow values that could come from instrument malfunction. The calculation of these outliers is inspired by the papers of Gudmundsson et al. (2018): daily streamflow values are flagged as outliers if values of log (Q+0.01) are larger or smaller than the mean value of log (Q+0.01) plus or minus 6 times the standard deviation of log (Q+0.01) computed for that calendar day over the entire series. The mean and standard deviation are computed for a 5-day window centered on the calendar day to ensure that a sufficient amount of data is considered. The log-transformation is used to account for the skewness of the distribution of daily streamflow values and 0.01 was added because the logarithm of zero is undefined.

To summarize, the quality of the 70 streamflow daily series is described using 12 flags (1, 2, 3, 4, A, B, C, D, E, F, H and I), as detailed in the Table 1. Using these criteria, along with visual analysis to identify anomalies (i.e. non-natural records that may be erroneous streamflow values or anthropogenic influences that can lead to misinterpretation of actual hydrological processes (Strohmenger et al., 2023)), 24 hydrometric stations were identified as "hydrologically relevant" from the 70 available.

Table 1 - Description of the 12 flags used.

| Flag | Description | Number of stations |
|---|---|---|
| 1 | station with at least one negative value (Q<0) | 0 |
| 2 | station with at least one outlier | 0 |
| 3 | station with at least one positive value (Q>0) that does not change over 10 time steps. | 60 |
| 4 | stations with less than 5 years of data available | 39 |
| A | Stations with well-established rating curve | 4 |
| B | Stations with very well-established rating curve for mean streamflow and medium to good for floods | 36 |
| C | Stations with rating curve passable | 6 |
| D | Stations with poor rating curve | 11 |
| E | stations located downstream of diversion channels or small dams used for irrigation | 4 |
| F | stations located downstream of resurgences or springs | 13 |
| H | Stations with manual water level reading | 50 |
| I | Stations with automatic water level reading | 11 |

### 3.1.2   Catchment boundaries and areas

The contours of the 24 catchments corresponding to the 24 selected hydrometric stations were delineated using the SRTM digital terrain model (Reuter et al., 2007) and the TauDEM algorithm (Tarboton et al., 2005).
The catchment areas calculated with TauDEM algorithm were compared with those reported in the "Hydrographic Bulletin" (areas estimated from U.S. Army maps). Table 1 in Appendix C presents the ratios and errors between the areas calculated with TauDEM and those in the hydrographic bulletins. The errors between the two areas are less than 10% for 18 of the 24 catchment areas (blue dots). However, significant errors were observed for 6 catchments (Q-045, Q-051, Q-056, Q-060, Q-061 and Q-065). Three factors
account for significant differences between the two areas:

1. The positions of some hydrometric stations were wrong in the archives. Their locations were corrected using additional information in the hydrographic bulletins (name of a bridge, main road, monuments, etc.). For example, the name of a bridge for station Q-056 (Pont Parois) and the name of the river for station Q-060 (Massacre river) were used to correct the station position

2. Due to the low resolution of the DEM, the river network generated with TauDEM algorithm may differ from the real river network, especially in plain areas near the estuaries. Hydrometric stations were

therefore relocated to match the stream generated by the TauDEM algorithm (stations Q-045, Q-065 and Q-051).

3. Three different stations (Q-053, Q-061 and Q-056) were associated to an upstream catchment area equal to 252 km².We supposed that this is an error in the areas of the hydrographic bulletins.

Hereafter, we will only use areas calculated with TauDEM algorithm and not areas noted in the paper archives. The geographic locations of the 24 selected hydrometric stations are shown as red dots in Figure 2.

## 3.2   Building catchment climate series

### 3.2.1   Rainfall

Three sources of rainfall data were used to build catchment scale rainfall series: (i) NOAA 20CR rainfall data, (ii) data from all available raingauges, and (iii) data from several possible combinations of raingauges.

1. NOAA 20CR rainfall data

Catchment-scale rainfall series were calculated as a weighted average of NOAA 20CR rainfall. The weights are proportional to the area of the NOAA pixel overlapping the catchment. The areas of most catchments are significantly smaller than the NOAA 20CR pixel. Thus, neighboring catchments located on the same NOAA grid cell will have the same rainfall series (see Figure 2).

2. Reference rainfall at the catchment scale

For each catchment, an initial rainfall series, called "reference rainfall" hereafter, was calculated as a weighted average of monthly rainfall data from Thiessen polygons (Croley and Hartmann, 1985; Han and Bray, 2006). Due to the high percentage of missing data in most rainfall series, the weights obtained from the Thiessen polygons are not the same for all time steps. For each time step, the weights are calculated using the raingauges with available data. The use of "reference rainfall", i.e. the use of all raingauges including those with a high percentage of missing data may introduce non-stationarity in the catchment-scale rainfall series and may not be "relevant" for rainfall-runoff modelling. The low density of raingauges and the high spatial variability of rainfall in Haiti (Moron et al., 2015) make it difficult to apply methods to estimate missing data (Benoit et al., 2022; Di Piazza et al., 2011; Oriani et al., 2020). Therefore, gap-filling methods were not used.

3. Multiple raingauge combinations

All possible raingauge combinations are calculated for each catchment (combination of 1, 2, 3,..., $n$ raingauges, where $n$ is the number of available raingauges). If a single raingauge is available, its data is used as the catchment scale rainfall series (weighting coefficient = 1). If there are multiple raingauges available, their weighting coefficients are calculated from the Thiessen polygons. Catchment scale rainfall series with no missing data were used for rainfall–runoff modeling.

4. Selection of the "relevant" raingauge for rainfall-runoff modelling

The performance of a rainfall-runoff model improves with a better description of the rainfall input (Andréassian et al., 2001). The GR2M monthly rainfall-runoff model was therefore used to determine, for each catchment and at the monthly timestep, the "relevant" raingauges in this study. NOAA 20CR rainfall series, reference rainfall series and multiple raingauge combinations are used as inputs to the GR2M model and relevant raingauges are defined as those providing the best model performance.

The first 3 years of data (early 1920 to late 1922) were used to initialize the model, and a split-sample test (Klemeš, 1986), commonly used in hydrology, was implemented. This practice consists in splitting a streamflow time series into two distinct subperiods P1 and P2, the first for calibration and the second for evaluation, and then exchanging these two subperiods. The two subperiods P1 and P2 are chosen so that they have the same available streamflow lengths. The combination of raingauges with the best KGE score in evaluation (average of the KGE in evaluation over the two subperiods) was considered as the most relevant for rainfall–runoff modeling.

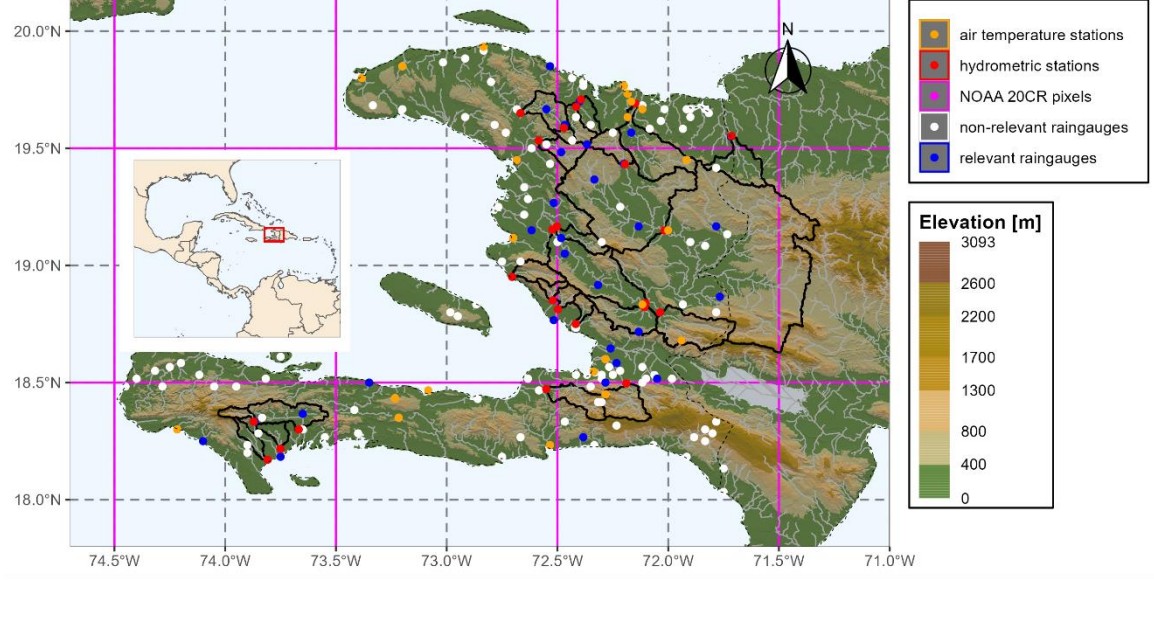

Figure 2 – Location of the 24 hydrometric stations used (red dots), the associated catchment contours (black solid lines), and the location of all raingauges with monthly data for the period 1920–1940 (white, orange, and blue dots). Raingauge stations with air temperature data are shown in orange. Raingauges considered relevant for hydrological modeling are shown in blue. NOAA 20CR pixels are shown in purple, the border between Haiti and the Dominican Republic is shown as a dashed black line, and the background topography is from the SRTM database.

### 3.2.2    Air temperature and potential evapotranspiration

The observed air temperature series are available at a monthly time step and are not available for the entire study period (1920–1940). In our context, continuous air temperature series are needed to estimate potential evapotranspiration (PET) series at the catchment scale. Because the air temperature series are incomplete, an annual average temperature was calculated for each station and used in the rainfall–runoff model.

Several studies have evaluated the impact of imperfect knowledge of air temperature data (using annual averages in our study) on the performance of rainfall–runoff models (Burnash, 1995; Fowler, 2002; Kribèche, 1994). The results converge to show that this source of uncertainty is the least important and that it can be largely compensated by the model during calibration. To verify this hypothesis, two complementary temperature databases (NOAA 20CR and BEST) were used as inputs to the GR2M model. The aim is to test whether the performance of the model (KGE score) is sensitive to differences in air temperature data.

1.    Using NOAA 20CR and BEST air temperature

Catchment air temperature series were computed at a daily time step for two temperature databases (NOAA 20CR and BEST) by taking the weighted average of pixels in the respective database (NOAA 20CR or BEST). The weights are proportional to the area of the NOAA 20CR or BEST pixel overlapping the catchment.

2.    Using available meteorological stations

Annual average temperature series were calculated for each catchment at the monthly time step using the observed (digitalized data) air temperatures. Daily air temperature series were then derived by interpolation from a second-degree polynomial. A similar study of interpolation of monthly temperature series to obtain daily temperatures was performed by Andréassian *et al.* (2004). Daily air temperature series at the catchment scale were calculated using the interpolated daily air temperature series and Thiessen polygons (Croley and

Hartmann, 1985; Han and Bray, 2006).

3. PET catchment series

The PET series are calculated using the formula of Oudin *et al.* (2005), which is based on air temperature. This formula was chosen for the calculation of PET for two main reasons. The other climate variables commonly used to calculate PET (wind speed, humidity, radiation, etc.) are unavailable, which justifies the

use of a formula based only on air temperature and extra-terrestrial radiation (which depends only on the Julian day and the latitude) in a context where data are scarce. Moreover, it is one of the most relevant approach for rainfall–runoff modeling compared to 27 models for calculating PET and has been tested on more than 300 catchments covering several climatic zones, including tropical zones (Oudin et al., 2005).

## 3.3 Water balance

The water balance was used as a complementary analytical tool to the GR2M model. The annual average water balance was presented in the form of a Turc–Budyko diagram, as described by Coron *et al.* (2015), for all 24 study catchments.

## 3.4 Simulation of monthly and daily streamflow series for the period 1920–1940

Three sets of parameters were used to simulate the streamflow series for each catchment during the period

1920–1940. The first two sets of parameters called P1 and P2, were obtained by calibration over the two subperiods using the catchment rainfall calculated from the relevant raingauges and the PET series calculated from the digitized temperature series, as described in section 3.2.1. The third set of parameters called P3 is obtained by calibration over the whole period 1920–1940 (the first 3 years being used to initialize the model). The GR2M model was used to simulate the monthly streamflow series for the 24 catchments studied, and the

GR4J model was used to simulate the daily streamflow series for 21 of the 24 catchments where daily rainfall data are available. Modeling was performed using the airGR package (Coron et al., 2017, 2020) and R software (R Core Team, 2022).

## 3.5 Calculation of catchment attributes

Similar to the CAMELS databases (Addor *et al.*, 2017, Alvarez-Garreton *et al.*, 2018, Chagas *et al.*, 2020,

Coxon *et al.*, 2020, Fowler *et al.*, 2021, Klingler *et al.*, 2021), a set of attributes that describes a broad range of low, moderate and high precipitation and streamflow characteristics were chosen to characterize the hydrological regime of each catchment. Thus, 49 attributes grouped into six classes (14 topographical attributes, 12 climatic attributes, 16 hydrological signatures, 2 land cover attributes, 4 geological attributes, and 1 hydrogeological attribute) were calculated (see Table C3 in Appendix C). Table 2 and Table 3

summarizes all the datasets used and produced in this study.

### 3.5.1 Location and topography attributes

Table C3 presents the six location indices that were calculated. Catchments are identified by the same codes as the hydrographic stations, in the format Q-XXX, where XXX ranges from 001 to 070 to identify the 70 hydrographic stations. The catchments have the same names as the hydrographic stations and are taken

from the hydrographic bulletins. The longitudes and latitudes of the outlets correspond to those of the hydrometric stations presented in section 3.1.2 (and includes coordinates modification). The longitudes and

latitudes of the catchment centroids were calculated based on the catchment contours delineated in section 3.1.2.

The topographic attributes include area, elevation, slope, catchment elongation, and drainage density. Catchment areas were calculated using the SRTM digital terrain model and the TauDEM algorithm (see section 3.1.2). Elevation is a key factor in hydrological processes as it influences many other catchment characteristics (Addor et al., 2017). Therefore, minimum and maximum elevations, standard deviations, hypsometric curves (empirical elevation distribution function) and average catchment slopes were calculated using the SRTM digital terrain model. The average slopes of the catchments were calculated using the SRTM digital terrain model and the algorithm of Horn (1981). The Gravelius index, which provides information on the elongation of the catchment and therefore influences the hydrograph, was calculated. The Gavelius index is defined as the ratio of the perimeter of the catchment to the circumference of a circle with the same area (Bendjoudi and Hubert, 2002). Finally, stream density, the ratio of the total of all stream segments to the area of the catchment, was calculated using the CNIGS (Centre National de l'Information Géospatiale in French i.e national center for geospatial information) river network shapefile. The stream density is influenced by the density of the hydrographic network and therefore by the permeability of the catchment.

### 3.5.2   Climatic attributes

The 12 climatic attributes (see Table C3) were determined using the monthly time series of rainfall, air temperature, and potential evapotranspiration at the catchment scale, which are available for the 1920-1940 period. These attributes include the "P_5_month", "T_5_month", and "PET_5_month" indices, representing the 5th percentile of rainfall, temperature, and potential evapotranspiration, as well as the "PMNA5" index (yearly minimum of monthly rainfall not exceeded once in 5 years), which represents the low values. The "P_mean", "T_mean" and "PET_mean" indices represent the mean values of precipitation, air temperature and potential evapotranspiration. The "P_95_month", "T_95_month" and "PET_95_month" indices represent the 95th percentile of precipitation, air temperature and potential evapotranspiration, while the "PMXA10" index (yearly maximum of monthly rainfall exceeded once in 10 years) represent the highest values. The return periods of the "PMNA5" and "PMXA10" indices were calculated using the generalized extreme value (Beirlant et al., 2004; Coles, 2001; Jenkinson, 1955). Finally, the aridity index was calculated. This index is the ratio of average rainfall to average evapotranspiration.

### 3.5.3   Hydrological signatures

The 16 hydrological attributes (see Table C3) for the 24 catchments studied were calculated using the observed and simulated streamflow time series available for the period 1920-1940 (see section 3.4). There are four indicators for each of the hydrological signatures (one indicator for observed streamflow and three indicators for simulated streamflow series). These attributes include "Q_5_month" (5th percentile of monthly data), "Q_5_day" (5th percentile of daily data), "QMNA5" (yearly minimum of monthly streamflow not exceeded once in 5 years), "low_q_freq" (frequency of low-flow days; < 0.2 times the mean daily flow) and "low_q_dur" (average duration of low-flow events; number of consecutive days < 0.2 times the mean daily flow) indices, which characterize the frequency, duration and magnitude of low flows. The Q_mean_month and Q_mean_day indices were used to characterize average flows at the daily and monthly time step. The "Q_95_month" (95th percentile of monthly data), "Q_95_day" (95th percentile of daily data), "QMXA10" (yearly maximum of monthly streamflow exceeded once in 10 years), "high_q_freq" (frequency of high-flow days; > 9 times the median daily flow), and "high_q_dur" (average duration of high-flow events; number of consecutive days > 9 times the median daily flow) indices were used to characterize the frequency, duration, and magnitude of high flows. Additionally, the runoff coefficients, baseflow index calculated according to the method proposed by Pelletier and Andréassian (2020), and parameters of the GR2M and GR4J models (see section 3.4) were provided.

### 3.5.4   Land cover

Land cover data for Haiti is provided by the CNIGS and is only available for two periods: 1995 and 1998.

Although the land cover classifications used in 1998 differ from those used in 1995, Figure *3* illustrates that
most of the woodland areas in 1995 were converted to cropland, grassland, or savannah in 1998. According
to the 1998 classification, medium-density cropland is the most dominant land use, accounting for a quarter
of the total the territory. High-density agro-forestry systems occupy 18%, high-density agricultural crops 17%,
savannah 7.3%, pasture with other uses 4.7%, wetlands 4.4%, rock outcrops and bare ground 1.8% and forest
1.25%. The area of other types of use is generally less than 1% of the territory.

Shapefile of land cover data (1995 and 1998) were cropped for each of the catchments studied. The proportion
of each land cover class occupied in the catchment was then calculated, corresponding to the two land cover
indices calculated in Simbi: "cover_95" (percentage of the catchment covered by each land cover class in
1995) and "cover_98" (percentage of the catchment covered by each land cover class in 1998).

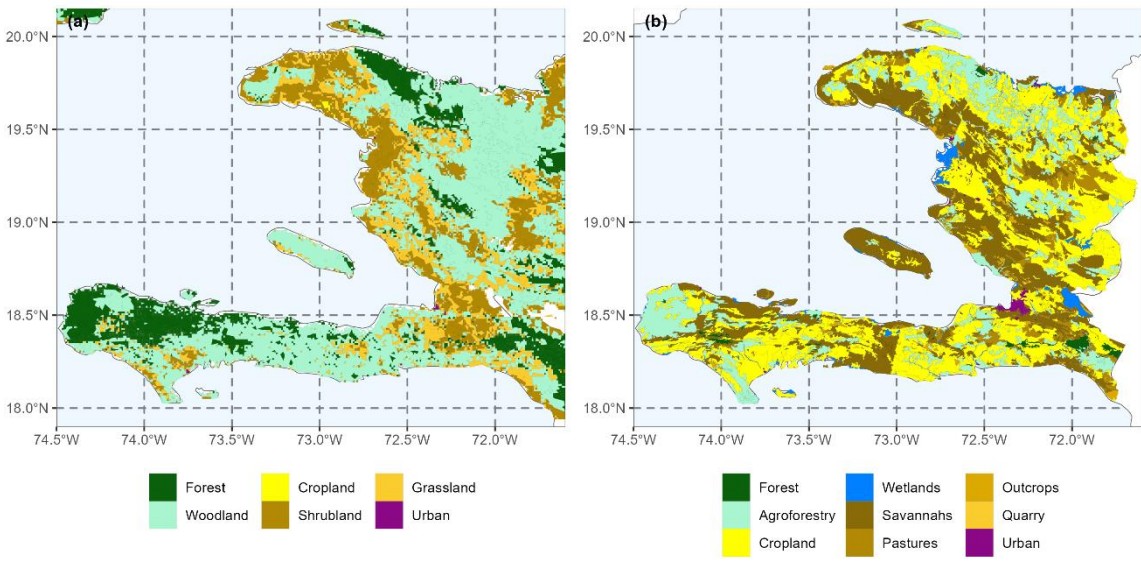

Figure 3 - (a) 1995 land cover map and (b) 1998 land cover map prodived by the CNIGS.

### 3.5.5    Geological attributes

The geological data provided by Butterlin (1960), Boisson and Pubellier (1987) and Terrier et al. (2014) have
been used and have been made available by the CNIGS. The most common lithology types in Haiti are
calcareous sedimentary rocks, followed by magmatic rocks (see Figure 4). The shapefile of lithology types
has been cropped for each of the catchment studied (Table C4 shows the list of geological classes). The
proportion of each lithology class in the catchment was calculated, corresponding to the "lithology" index. The
proportion of carbonate rocks, sedimentary rocks and magmatic rocks has been calculated for each of the
catchment and corresponds to the "Carb_Rocks_Perc", "Sedim_Perc" and "Magma_Perc" indices.

### 3.5.6    Aquifer attributes

The aquifer data were produced by the MARNDR (Ministry of Agriculture, Natural Resources and Rural
Development) in the 1990s have been used and have been made available by the CNIGS. Carbonate aquifers
are the most widespread in Haiti, consist of carbonate rocks, mainly limestone and marl, and cover 53% of
Haiti's surface area, of which karstic aquifers account for 18%. Crystalline formations, mainly magmatic rocks,
account for 17%, alluvial aquifers for 16% and low-permeability sedimentary formations for 13%. Figure 4
shows the spatial distribution of the different aquifer classes and Table C4Table shows the list of aquifer
classes. The shapefile of aquifer classes has been cropped for each of the catchments studied. The proportion
of each class in a catchment was then calculated, corresponding to the "aquifer" index.

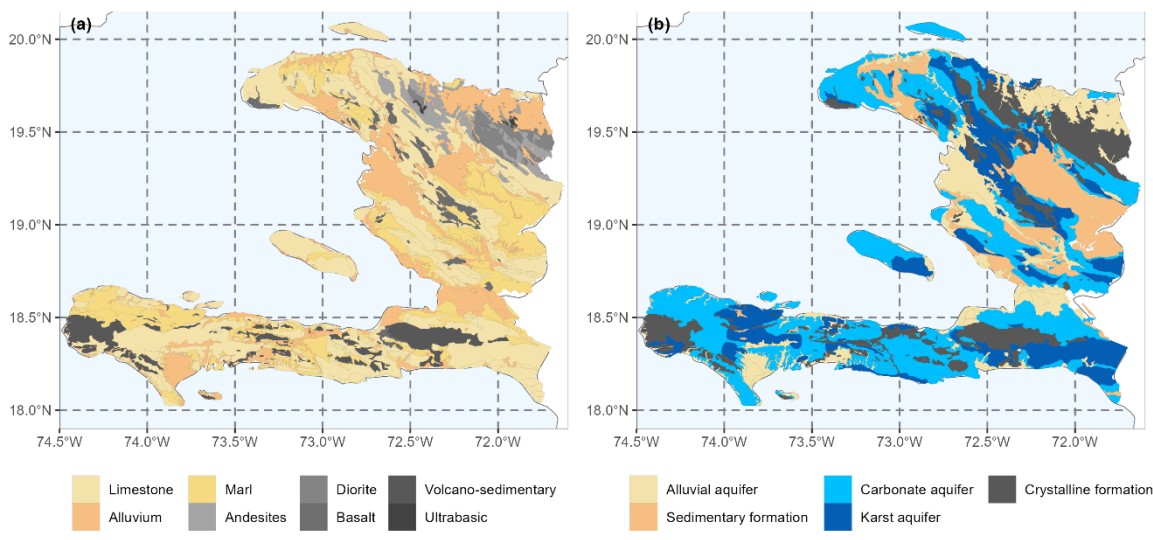

Figure 4 - (a) lithological classes are represented by light colors for sedimentary rocks and shades of grey for magmatic rocks. (b) aquifer classes are represented by light colors for alluvial aquifers, blue colors for carbonate aquifers, and grey for crystalline aquifers.

Table 2 - Summary of the datasets used in this study.

| Datasets | Source | Period of data availability |
|---|---|---|
| 156 monthly rainfall series | Moron et al., 2015 | 1905-2005 |
| 70 daily streamflow series. | BVH project | 1920-1940 |
| Paper archives contain daily rainfall series | BHS | 1920-1940 |
| Paper archives contain monthly air temperature series | BHS | 1920-1940 |
| NOAA 2OCR rainfall and air temperature daily database | Slivinski et al., 2019 | Twentieth Century |
| BEST air temperature database | Rohde et al., 2013 | Since 1753 |
| SRTM DEM with a resolution of 90 m | Reuter et al., 2007 | - |
| Shapefile of lithological classes on Haiti | CNIGS | - |
| Shapefile of aquifer classes on Haiti | CNIGS | - |
| Shapefile of land cover classes on Haiti | CNIGS | - |
| Shapefile of Haitian stream network | CNIGS | - |

Table 3 – Summary of the datasets produced in this study.

| Datasets | Period of data availability |
|---|---|
| Digitization of 59 daily rainfall series | 1920-1940 |
| Digitization of 23 monthly air temperature series | 1920-1940 |
| Rainfall, air temperature and PET series at catchment scale and at daily and monthly time steps for 24 catchments studied. | 1920-1940 |
| Simulated streamflow series at daily and monthly time steps for 24 catchment studied. | 1920-1940 |
| 49 attributes for each of the 24 catchment areas studied | - |

# 4 Results

## 4.1 Impact of air temperature and PET series on rainfall–runoff modeling

Figure 5 shows (i) the relationship between digitized air temperatures (BHS) and the 20CR and BEST reanalyses ((a) to (c)) as well as (ii) the performance (KGE score) and parameters of the GR2M model ((d) to (f)) using the three air temperature databases to compute PET series. The BEST database overestimates the
460 mean air temperature (symbolized by the red dots in the boxplots in Figure 5 (a)) and 20CR has difficulty representing temperatures below 20°C and over 28°C. The low dispersion (Figure 5 (b)) of 20CR and BEST may be due to a spatial averaging effect at the scale of the grid boxes which are large for the study area (1° for both). In addition, there is no linear correlation between the two temperature databases, and the 20CR data poorly represents the seasonal temperature variability in Haiti (Figure 5 (c)).

Although there is no clear correlation between the digitized and reanalyses temperatures, the KGE values (KGE in the evaluation for the two subperiods) obtained with the three air temperature databases are very similar for most of the catchments (Figure 5 (d)). This shows that the GR2M model, through its two parameters and especially the X2 parameter (Figure 5 (f)), has the ability to absorb the potential biases associated with the air temperature data.

Thus, the three temperature databases could be used a priori for rainfall–runoff modeling, as the model parameters absorb the associated biases. However, since the reanalysis databases do not represent temperature well at the catchment scale, they will not be used in the remainder of this study. Therefore, the digitized temperatures will be used to build the Simbi database.

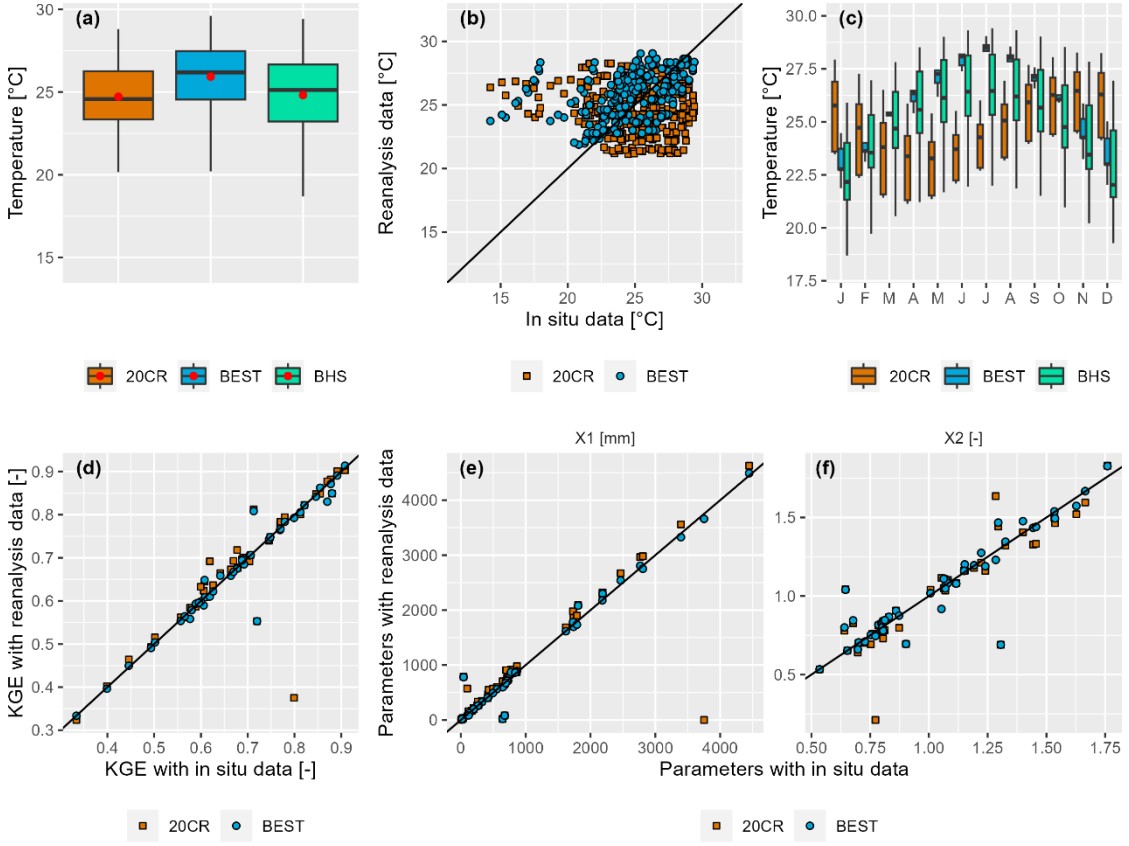

Figure 5 – (a) monthly air temperatures at catchment scale for the three datasets (20CR, BEST and BHS data observed in situ). (b) monthly air temperatures at catchment scale from in situ data versus reanalyses (20CR and BEST). (c) seasonal temperature variability as a boxplot (each boxplot represents monthly temperatures for all catchments). (d) KGE values in the evaluation for the two subperiods. (e) et (f) GR2M parameters X1 and X2 obtained with in situ air temperatures versus those obtained with reanalyses air temperatures.

## 4.2    Selection of relevant raingauges

### 4.2.1    GR2M performance analysis

Reference rainfall and rainfall from all possible combinations of raingauges were calculated at the catchment scale as described in section 3.2.1. Table C2 in Appendix C presents the number of raingauges used to calculate the reference rainfall, the number of combinations, and the most relevant raingauges for rainfall–runoff modeling for each of the catchments.

Figure 6 shows the summary of the GR2M KGE scores and its three components obtained with NOAA 20CR rainfall, reference rainfall, and relevant raingauge combination. The lowest KGE scores are obtained with NOAA 20CR rainfall, highlighting the limitations of this rainfall database for rainfall–runoff modeling in Haiti, and the need to use observed data rather than reanalyses. There is also a clear improvement in KGE values when using the relevant raingauges compared to the reference raingauges. Nevertheless, some catchments have poor KGE scores in evaluation, despite the use of relevant raingauges. Among the three components of the KGE, the correlation coefficient ($r$) contributes most to the improvement in model performance through the use of ground-based rainfall data. Indeed, there is a weak correlation between the simulated and observed streamflow obtained with NOAA 20CR rainfall data, and this correlation is greatly improved by using observed in situ rainfall data (reference raingauges and relevant raingauges). On the other hand, the coefficients $\alpha$ and $\beta$, which represent variability and bias, respectively, contribute most to the improvement of the model

performance using the relevant raingauges compared to the reference raingauges. The values of these coefficients are much more centered around the optimal value of 1 for the relevant raingauges, while they are more scattered for the reference raingauges.

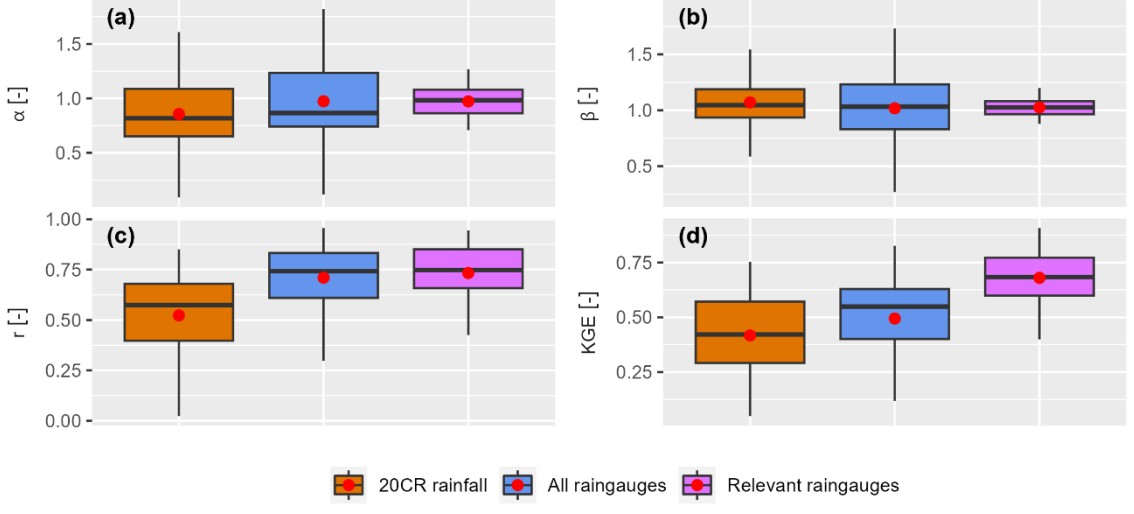

Figure 6 – (a) to (d) summary of GR2M KGE scores and its three components obtained in evaluation with NOAA 20CR rainfall, reference rainfall, and relevant raingauges combination for the 24 studied catchments.

Figure 7 shows the spatial distribution of GR2M KGE scores in evaluation using the three rainfall databases. As discussed earlier, the KGE scores in evaluation with 20CR data are low, with only five catchments having
KGE scores over 0.60. The performance improved for 21 catchments with the relevant raingauge combination, and no improvement was achieved for three catchments only: the catchments of Tumbe at Passe Fine (Q-044), Rivière du Sud at Camp-Pérrin (Q-008), and Coujol at Proby (Q-006). Two of these three catchments (Q-044 and Q-008) were already performing relatively well, with average KGE in the evaluation over 0.60, and the use of the relevant raingauge combinations did not improve their performance further. Despite the use of
relevant raingauges, four catchments have KGE values below 0.50, two of which have negative or near-zero KGE values: the Trois Rivières at Plaisance catchment (Q-051) and the Montrouis at Pont Toussaint catchment (Q-058).

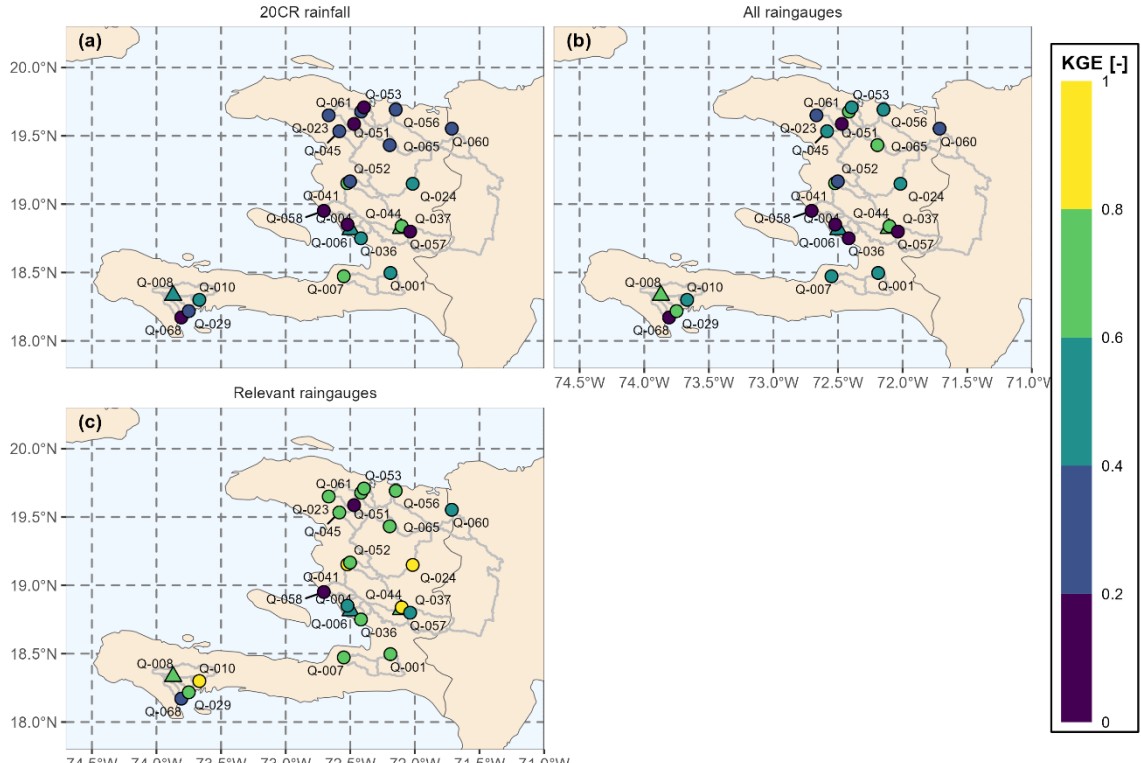

Figure 7 - Spatial distribution of the average of the two KGE values obtained with GR2M in evaluation for the two subperiods. KGE values are calculated using (a) 20CR rainfall data, (b) reference raingauges (all raingauges) and (c) relevant raingauges. Dots represent catchments where model performance was improved by using the relevant raingauge combinations, triangles represent catchments where model performance was not improved by using the relevant raingauge combinations.

### 4.2.2   Analysis of GR2M parameters

In *Figure 8*, the influence of the relevant raingauge combinations on the stability of the model parameters is evaluated. The ratios of the parameters calibrated over the two calibration subperiods were plotted as a boxplot for the reference and relevant raingauge combinations. The results showed that the relevant raingauge combinations led to more stable X1 and X2 parameters (ratio close to 1). Overall, the relevant raingauges led to a better performance and stability of the model parameters.

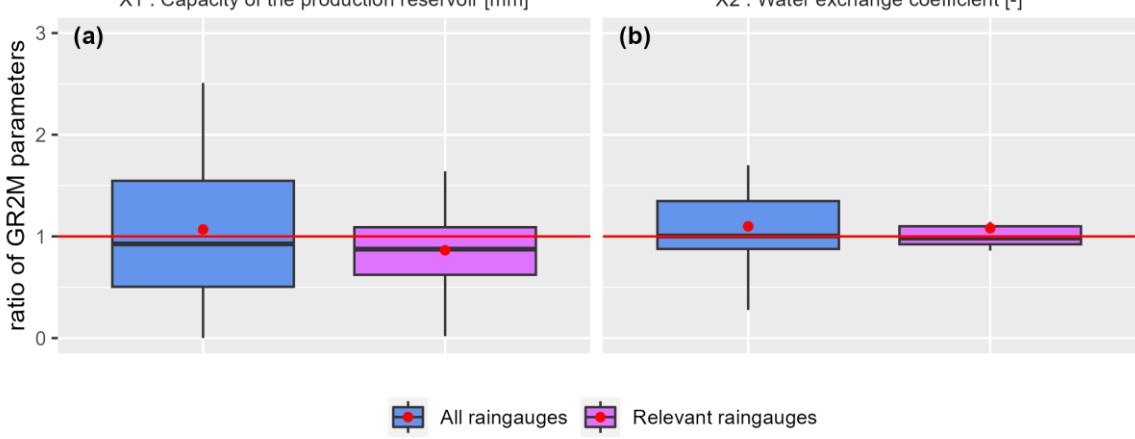

Figure 8 – Ratio of the GR2M calibrated parameters X1 (a) and X2 (b) over the two subperiods for the reference and relevant raingauge combinations. The red line represents the optimal ratio (r=1), while the red dot represents the mean value of the distribution.

### 4.2.3    Characteristics of relevant raingauge combinations

Figure 9 shows that the raingauges used for the relevant raingauge combinations are those located at low elevations and with the longest data series. The relatively low percentage of missing data from the relevant raingauges ensured better model stability (see section 4.2.2) and contributed to the improvement in the model performance, especially by reducing the biases between simulated and observed streamflow (improvement in $\alpha$ and $\beta$ parameters; see section 4.2.1). Raingauges at higher elevations are more difficult to access and
are the least maintained, and therefore have very high percentages of missing data (raingauges with less than 10 years of data). However, the model tends to discard raingauges with high percentages of missing data, which is why the retained/selected raingauges are generally located at lower elevations. There is no clear trend of monthly rainfall in the selection of relevant raingauges. However, some very wet raingauges (rainfall totals over 180 mm/month) were selected as relevant raingauges.

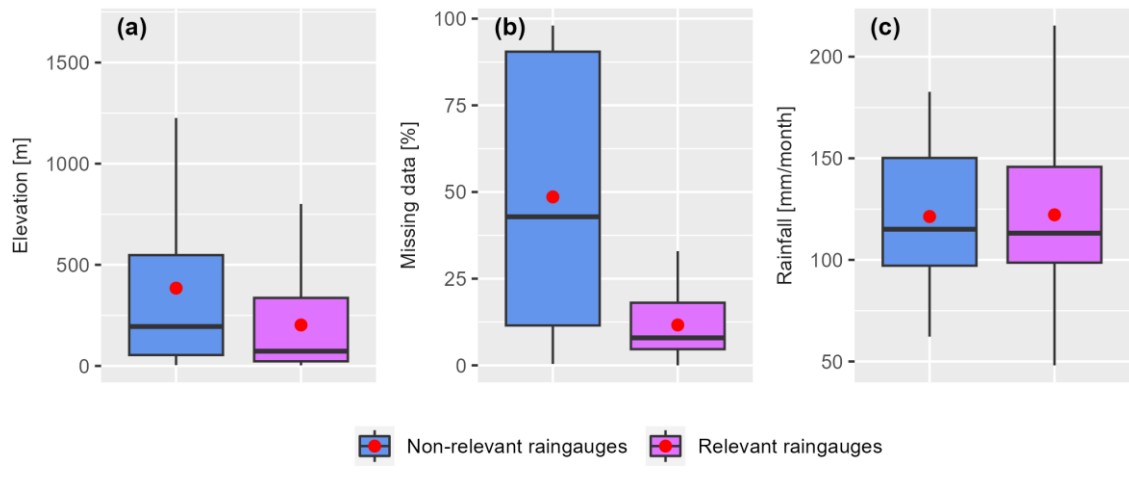

Figure 9 – (a) distribution of raingauges elevations, (b) percentage of missing data, and (c) monthly rainfall for relevant (40) and non-relevant (21) raingauges.

## 4.3  Water balance

The average annual water balance, in the form of a Turc–Budyko diagram, was used as another diagnostic tool to verify the hydroclimatic consistency of the assembled dataset. The results, presented in Figure *10*, show that the studied catchments correspond to conservative catchments (points located in the white part of the graph, i.e., Q < P and P - Q < PET), except for the catchments of Rivière du Sud at Camp-Perrin (Q-008) and Rivière Grise (Q-001). More than 90% of the Q-008 catchment is on a calcareous geological formation, and part of the catchment is also affected by karstic aquifers. Therefore, there may be a contribution of water from neighboring catchments that justifies such a high Q/P ratio, but no such study has been conducted to confirm or refute this hypothesis. The interpretation of the results for the Q-001 catchment is more difficult, as it may be related to the choice of relevant raingauge combinations for this catchment, or to a real exchange of streamflow with the neighboring catchments, or to a mixture of both. The water balances obtained with the relevant raingauge combinations are shifted to the lower right (blue circles and triangles). This is related to the fact that some of the raingauges used are very wet (see section 4.2.3) and therefore increase the rainfall at the catchment scale. No clear trend was observed between the water balances obtained with observed streamflow and those obtained with simulated streamflow.

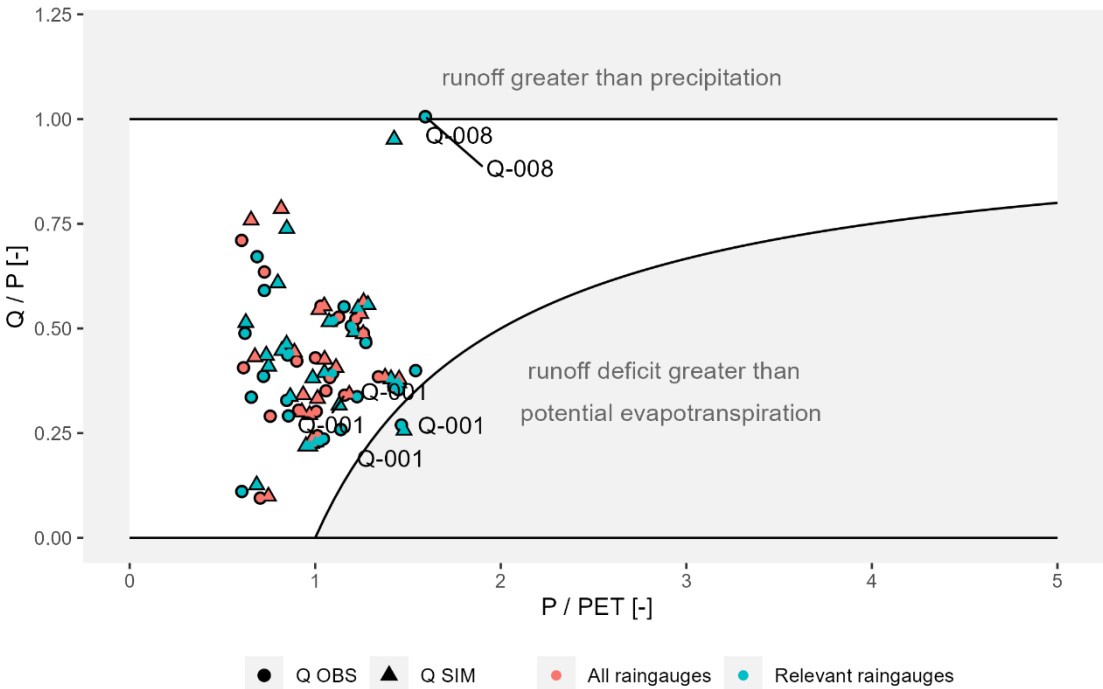

Figure 10 – Average annual water balance in the form of a Turc–Budyko diagram for all 24 catchments. The reference raingauges are shown in red and the relevant raingauges are shown in blue. Observed streamflows are shown as circles and simulated streamflows with parameters calculated over the whole period of available data are shown as triangles.

## 4.4  Performance of the rainfall–runoff models

Three sets of parameters (see section 3.4) were used to simulate three sets of monthly streamflow for each of the 24 catchments using the GR2M rainfall–runoff model, forced by the relevant raingauge combinations and the PET calculated with digitized air temperatures. The results, presented in Figure 11 (a), show that the KGE scores have a median value of 0.75 in calibration and 0.67 in evaluation.

The relevant raingauges have daily data for 21 of the 24 catchments. Therefore, daily streamflow series were simulated by the GR4J model for these 21 catchments. The KGE scores have a median value of 0.57 in calibration and 0.44 in evaluation (Figure 11, (b)). The daily rainfall data used as input to GR4J may partly

explain the low KGE values obtained. Indeed, raingauges with high percentages of missing data led to instability and poor performance of the GR2M model in most catchments, which required searching for relevant raingauges to improve the stability and performance of the model at the monthly time step (see section 4.2). However, there is a higher percentage of missing data in the available daily rainfall data than in the monthly data. Furthermore, the limited availability of daily data makes it difficult to improve the performance of the model at the daily time step.

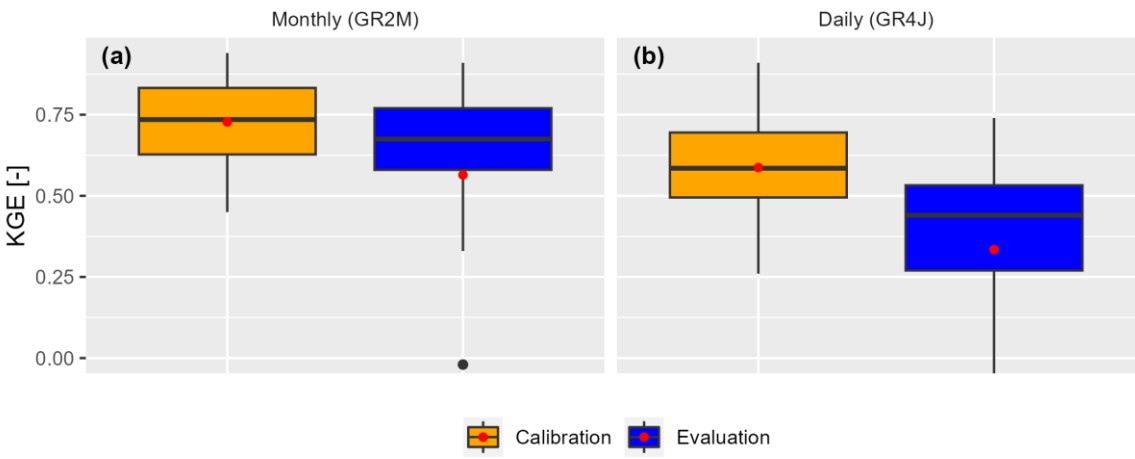

Figure 11 – Synthesis of KGE scores in calibration and evaluation at monthly (a) and daily time steps.

## 4.5   Catchment attributes

The 49 catchment attributes were calculated as described in section 3.5 and Table C3. Results for all attributes are not presented in this paper. Only some climate indices and hydrological signatures are presented.

### 4.5.1   Hydrological signature at the monthly time step

The observed and simulated mean annual streamflows from GR2M are illustrated in Figure 12. The results show that streamflow is higher in the southwest and north of Haiti and lower in the central part. However, the Q-008 catchment has a significantly higher mean annual streamflow than its three neighboring catchments (Q-010, Q-068 and Q-029). As shown in section 4.3, over 90% of the Q-008 catchment is situated on a calcareous geological formation, 40% of which is on karstic aquifers. Therefore, it is probable that an influx of water from neighboring catchments is accountable for such a high mean annual streamflow. Nevertheless, no study has been conducted to confirm or dispute this hypothesis. The simulated streamflow represents well the spatial pattern of the observed streamflow and gives good estimates of the observed mean annual streamflow.

Figure 13 shows the rainfall and streamflow regimes for the studied catchments. The results show a bi-modal rainfall/streamflow regime with two seasons of heavy rainfall/streamflow: the first season occurring around May and the second season between September and November, which corresponds to the cyclonic season. Rainfall is highly variable during the cyclonic season, with relatively heavy rainfall recorded in some catchments. The simulated streamflow represents well the seasonality of the observed streamflow (see Figure 13). However, simulated streamflows overestimate the observed values in May and underestimate them in November. In addition, the simulated streamflows slightly overestimate the low values in January. A time lag has been observed between the peak rainfall in October and the peak flow in November. This lag can be explained by soil saturation. The second season of heavy rainfall in Haiti, from September to November, gradually moistens the soil until it is saturated. Although the rainfall in November is relatively lower than in October, the streamflows in November are generally higher due to soil saturation. However, this hypothesis requires further investigation in future study.

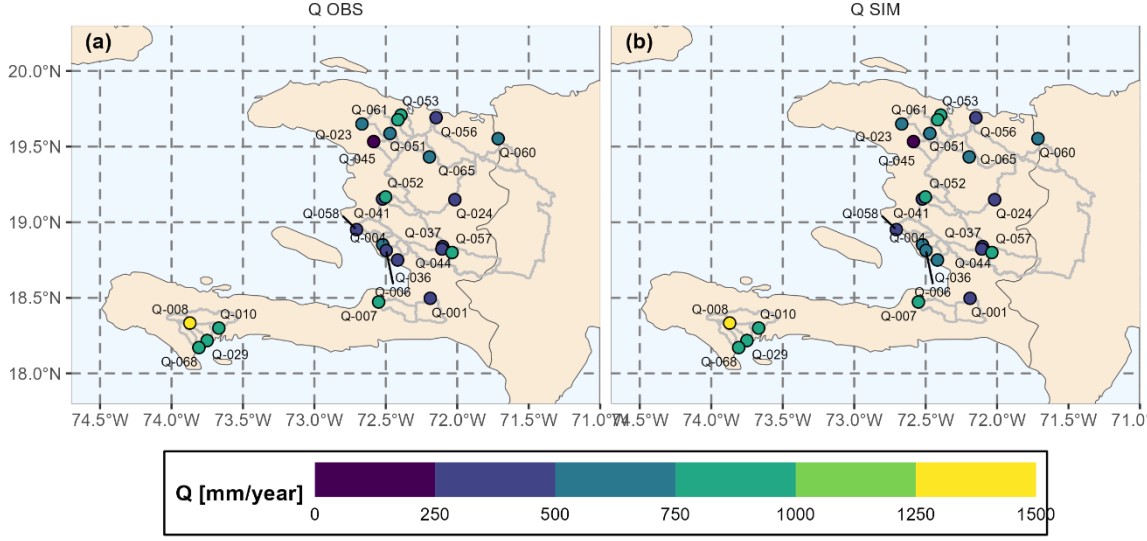

Figure 12 – Spatial distribution of observed mean annual streamflows (a) and simulated streamflows (b) with the GR2M parameters calculated over the entire period of available data.

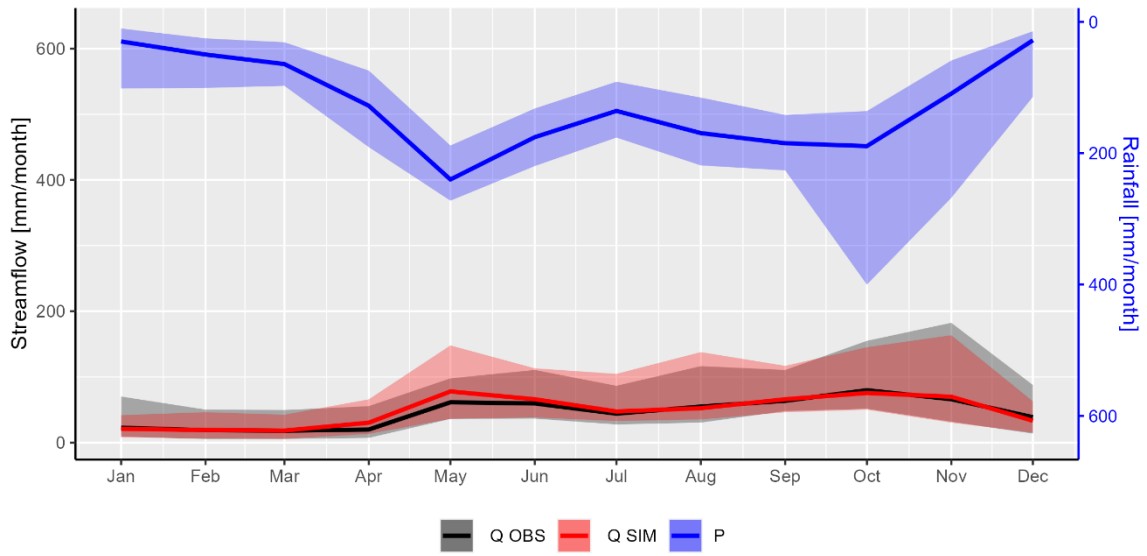

Figure 13 – Seasonality of rainfall (obtained by combining the relevant raingauges) in blue, observed streamflow in black, and simulated streamflow with the parameters calculated over the entire period of available data in red. The ribbons Values ranges have been estimated using represent the range of values between the 10th and 90th percentiles, while the thick line represents the median values for the 24 catchments studied.

The aridity indices and runoff coefficients are presented in Figure 14. The aridity indices show the same spatial pattern as the mean annual streamflow (Figure 12), i.e., they are greater than 1 in the central part of Haiti (arid zone) and lower in the southwest and north (humid zone).

The runoff coefficients are approximately 0.35 for catchments in the central zone and approximately 0.5 in the southwest and north of Haiti. The South River catchment at Camp-Perrin (Q-008), discussed above, has a runoff coefficient greater than 1, meaning that runoff is greater than rainfall. This high runoff coefficient can be explained by the presence of karst aquifers in the Q-008 catchment.

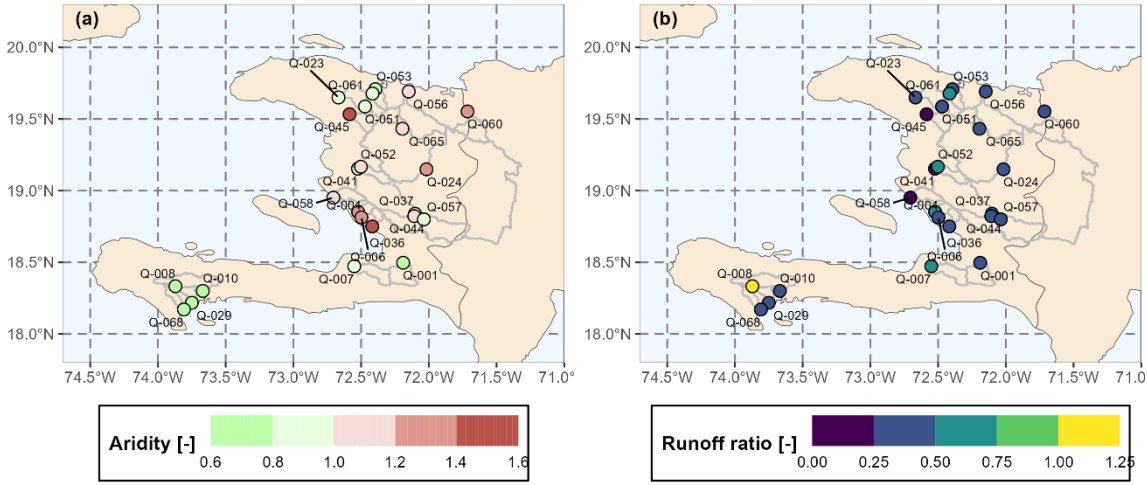


Figure 14 – (a) Aridity index calculated from rainfall series from relevant raingauges. (b) Runoff coefficient calculated from observed streamflow series on the right.

### 4.5.2   Hydrological signatures at daily time step

The hydrological attributes of the simulated and observed streamflows for the 21 selected catchments are summarized in Figure 15. The results show that the simulated streamflows are able to represent average daily streamflow well, underestimate low streamflow (5% quantile), and overestimate high streamflow (95% quantile) and baseflow indices (Pelletier and Andréassian, 2020). These overestimates of high streamflow and underestimates of low streamflow result in increased frequencies and durations of simulated high and low streamflow relative to observed streamflow. This poor representation of simulated high and low streamflow is a consequence of the poor performance of the GR4J model for most catchments.



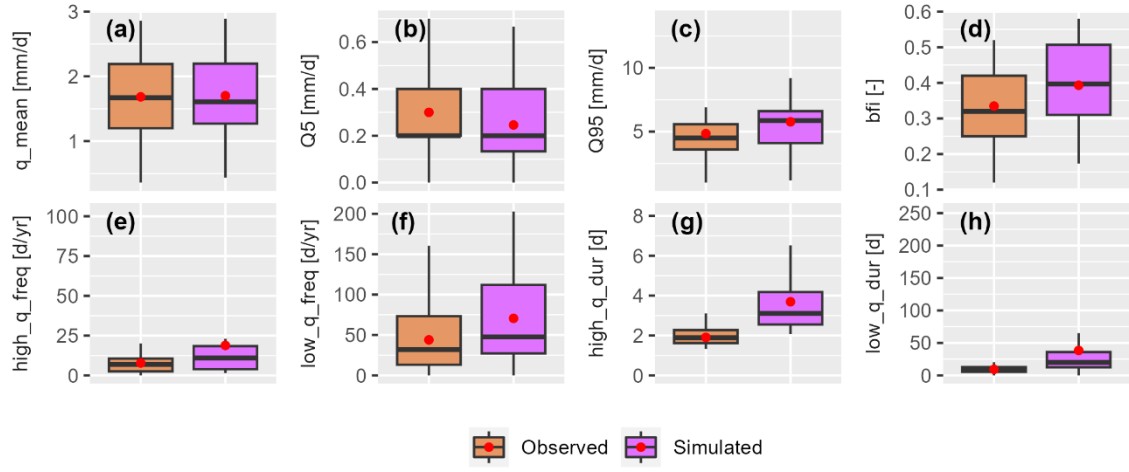

Figure 15 – Summary of hydrological signatures at the daily time step for observed and simulated streamflow and for 21 catchments. From (a) to (h): daily mean streamflow, 5% quantiles, 95% quantiles, baseflow, high streamflow frequencies, low streamflow frequencies, high streamflow durations, and low streamflow durations.


## 4.6   Graphical summary sheets of the Simbi database

The main catchment characteristics are summarized in sheets. These summary sheets have been inspired by those prepared by the catchment hydrology research group at INRAE (Brigode et al., 2020). An example is shown in Figure 16 describing the main characteristics of the Cavaillon catchment, which was studied

several times after Hurricane Matthew (Mathieu, 2023; Joseph, 2019; Joseph et al., 2018). This catchment

has an area of 320 km², half of which is at an elevation above 250 m, with a slope greater than 10°, and overlies a karst aquifer. During the rainy season (April–November), the catchment receives more than 150 mm/month of rainfall and more than 280 mm/month during the peak rainfall in May and November. Streamflows can reach 100 mm/month during May–June and October–November. Simulated streamflows underestimate maximum annual flows with a return period of less than 10 years and overestimate flows

beyond 10 years. The generalized extreme value (Beirlant et al., 2004; Coles, 2001; Jenkinson, 1955) and the distribution of annual values (precipitation, PET, air temperature or streamflow) were used to estimate values for multiple return periods. Simulated streamflows also underestimate low flows (annual minimums). Thus, during flood periods, we can expect daily streamflows of several hundred mm, and conversely, during dry periods, streamflows can be on the order of 10 mm/month.


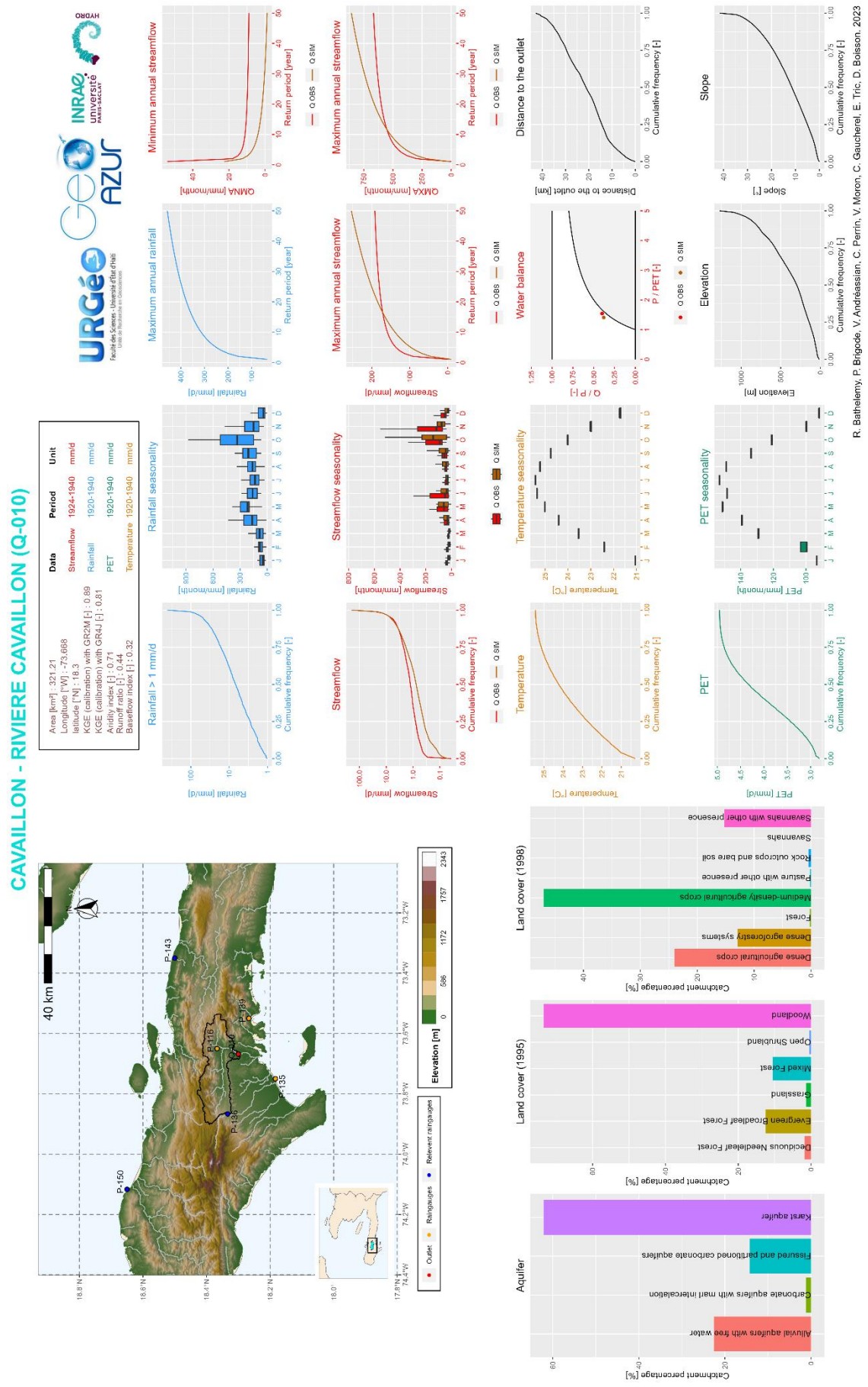

Figure 16- Summary sheet of the characteristics of the Cavaillon catchment.

# 5   Uncertainties

This section discusses the main sources of uncertainty associated with the Simbi database. These uncertainties can be classified into four main types:

1.   The Simbi database contains historical data, which may be prone to errors due to factors such as the used equipment, methods employed to measure flows, and the establishment of rating curves. For most streams, water levels were measured manually by reading a vertical scale placed on one 660   of the banks of the stream two or three times a day. Over time, 12 automatic recorders have been installed on 11 rivers, providing automatic and continuous readings of water levels on these streams. The metadata indicates the quality of the rating curves and the stations where the automatic recorders have been installed.

2.   The historical data was originally in paper format and has been digitized. Despite quality control tests, 665   uncertainties remain regarding the digitization of paper archives.

3.   The raingauges identified as relevant for hydrological modelling in this article depend on the use of a rainfall-runoff model. Different methodologies or rainfall-runoff models may produce different results and thus different catchment-scale precipitation forcing.

4.   The simulated streamflows are dependent on the rainfall-runoff models that are used and may differ 670   if other models are used. It is important to note that, especially at the daily time step, the KGE obtained for some catchments are poor (KGE<0.5).

# 6   Data availability

The Simbi database is freely available for download at: https://doi.org/10.23708/02POK6 (Bathelemy et al., 2023). The SIMBI_README.txt file contains a description of the database and the organization of the various 675   files and folders. Missing data in the Simbi database are indicated by -9999.

# 7   Conclusion and perspectives

To the best of our knowledge, the hydro-meteorological database presented in this article represents the first open access, and exhaustively documented hydro-meteorological dataset for Haiti. This database, called "Simbi", contains station observations and catchment-scale data. The station observations contain:

1.   59 daily rainfall series available from 1920 to 1940,

2.   156 monthly rainfall series available from 1905 to 2005,

3.   70 daily streamflow series available from 1920 to 1940,

4.   23 monthly air temperature series available from 1926 to 1939.

The data at the catchments scale contain:

1.   Climate forcings (precipitation, air temperature and potential evapotranspiration) a both monthly and daily timestep for 24 and 21 catchments, respectively.

2.   Simulated monthly streamflow series for 24 catchments and simulated daily streamflow series for 21 catchments using three sets of parameters (three simulated streamflow series per catchment) from the GR2M monthly and GR4J daily rainfall–runoff models, and

3.   A set of indices that describe a wide range of low, moderate, and heavy rainfall and streamflow characteristics to characterize the hydrological regime and water resources management applications.

The Simbi database highlights the spatial variability of Haiti's hydrological conditions. The central part of Haiti is associated with relatively low streamflow and high drought coefficients. The southwest is associated with
relatively high streamflow. In fact, large floods are more frequent in these areas (Terrier *et al.*, 2017). No clear trend was observed in the north. The simulated monthly streamflows perform well in representing average streamflow and their spatial variability. However, the model is less effective at the daily time step (KGE score in evaluation is below 0.5 for most catchments). This results in poor representation of the frequency or number of consecutive days with high and low streamflow. This may be due to a combination of the quality of the data
used and the calcareous geological formations that can create non-conservative catchments that are difficult to model.

Our database can be considered a starting point for any hydroclimatic study in Haiti, since it gathers, in addition to the simulated data, all the hydroclimatic data available in Haiti over several years. The database could contribute to better knowledge of the hydro-climatology in the twentieth century, and to study the
evolution of the climate in Haiti for better adaptation to climate change. Frequency analysis methods can be utilized to estimate flood return periods. The accessibility of streamflow data allows for the possibility of various rainfall-runoff modeling approaches to be applied. Overall, this hydrological database will contribute to a better understanding of hydrological risk in Haiti. The database will be regularly updated by integrating the historical archives that will later be digitized, making it the most complete hydrological database in Haiti. However, Simbi
is associated with several sources of uncertainty, including data quality (historical data), digitization of paper archives, identification of relevant raingauges, and rainfall-runoff models. It is important to consider these uncertainties when using Simbi.

## Competing interests

The authors declare that they have no conflict of interest.

## Author contribution

Conceptualization and methodology: RB, PB, VA, CP. Data curation: RB, VM, CG. Original draft preparation RB. Review and editing: all authors.

## Acknowledgments

The authors thank the Bibliothèque Haïtienne des Spiritains (BHS) for providing the paper archive of daily rainfall data and the BVH project for providing the daily streamflow series. Special thanks to the students Eddy-Terson François, Douninio Jeanite, Appollon Jean Philippe, and John Claury Ménélas of the Université d'Etat d'Haïti who contributed to digitizing the daily rainfall data and to the students Kathleen Gerarduzzi, Camille Morillon, and Alexandre Antony of the Université Côte d'Azur who digitized the monthly air
temperature data. Olivier Delaigue's work inspired the design of our files, and we extend our thanks to him. Thanks to Isabella Athanassiou for editing the English version of the manuscript. The authors thank the three reviewers and the editor who provided constructive comments on an earlier version of the manuscript, which helped clarify the text.

The authors also thank the CLIMEXHA project (Anticipating Extreme CLIMATE events over HAITI for a
sustainable development) and *Fonds pour la Recherche et le Développement de la BRH (Haiti)*, which contributed financially to the digitization of the daily rainfall data. This study is part of Ralph Bathelemy's Ph.D. thesis funded by the Anténor Firmin grant from the French Embassy in Haiti, the Institut de Recherche pour le Développement (IRD) through the ARTS grant, and the CARIBACT International Mixed Laboratory.

**Appendix A**

A verification of the two rainfall databases used was performed by comparing the monthly totals of the digitized daily rainfall series with the monthly rainfall database created by Moron *et al.* (2015). For months where the monthly totals of the two databases differed, a re-verification of the digitized daily rainfall series was carried out, which improved the quality of the digitized daily rainfall data. For some months and stations, the rainfall data produced by Moron *et al.* (2015) were erroneous. The errors in the Moron *et al.* (2015) data are generally of five types:

1. A data entry error during the digitization of this monthly data.

2. Data from some months are confused with data from another station with similar names (e.g., St. Louis du Nord and St. Louis du Sud, Verrettes and Fonds Verrettes), which are often not geographically close.

3. Elimination of some extreme values, thinking that they were input errors. For example, the rainfall was in fact 1196.9 mm at Camp-Perrin (P-136) in October 1933, but the Moron *et al.* (2015) database stated 196.9 mm.

4. Error in calculating monthly totals. In fact, at the end of each month, the monthly rainfall totals were calculated by the raingauge managers, and sometimes there were errors in calculating the monthly totals. However, it is these monthly totals that were used to create the Moron *et al.* (2015) data.

5. Mixing of data from stations located in the same city. Initially, all the raingauges were managed by the observatory of the Petit Séminaire Collège St. Martial, and these raingauges were named after the town in which they were installed. Around 1928, public works began to install stations in the same towns as the first stations. This sometimes led to confusion between neighboring stations. For example, the 1920–1930 data for the Hinche station (P-065) are from the observatory station, and the 1931–1940 data are from the public works station. To avoid confusion, only the observatory stations were used in our study because they are more numerous and contain the longest data series.

## Appendix B

GR2M (Mouelhi et al., 2006) is a monthly lumped rainfall–runoff model. Its structure (see Figure B1) combines a production store and a routing store to simulate the hydrological behavior of the catchment. The model has two parameters to optimize during calibration:

1. X1: the production store maximal capacity [mm],

2. X2: the catchment water exchange coefficient [-],

GR4J (Perrin et al., 2003) is a daily lumped rainfall–runoff model. Its structure (see Figure B1) combines a production store and a routing store and unit to simulate the hydrological behavior of the catchment. The model has four parameters to optimize during calibration:

3. X1: the production store maximal capacity [mm],

4. X2: the catchment water exchange coefficient [mm/day],

5. X3: the 1-day maximal capacity of the routing store [mm],

6. X4: the HU1 unit hydrograph time base [days].

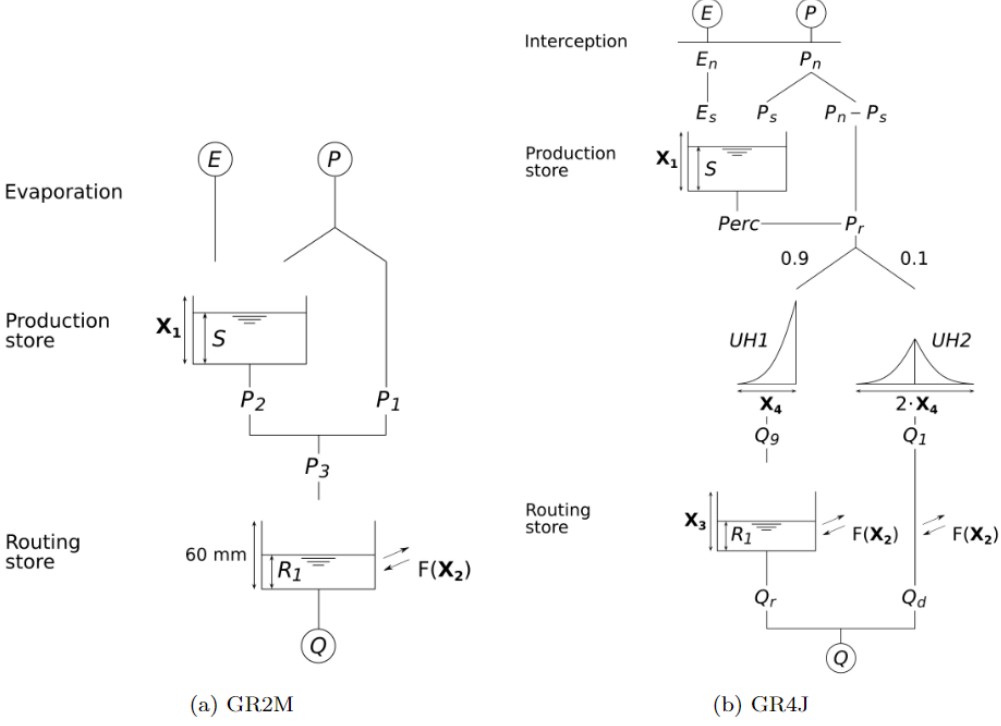

Figure B1 - Diagram of GR2M and GR4J models.

 **Appendix C**

Table C1 - Ratio and error between the catchment areas calculated with TauDEM and those of the hydrographic bulletins.

| Catchment code | Ratio [-] | Error [%] |
|---|---|---|
| Q-001 | 0.99 | 1 |
| Q-004 | 0.97 | 2 |
| Q-006 | 0.95 | 5 |
| Q-007 | 1.01 | 1 |
| Q-008 | 1.01 | 1 |
| Q-010 | 1.03 | 3 |
| Q-023 | 1.01 | 1 |
| Q-024 | 1.05 | 5 |
| Q-029 | 1.08 | 8 |
| Q-036 | 0.92 | 8 |

| Q-037 | 1.00 | 0 |
|---|---|---|
| Q-041 | 0.99 | 1 |
| Q-044 | 1.02 | 2 |
| Q-045 | 1.50 | 50 |
| Q-051 | 0.88 | 12 |
| Q-052 | 1.04 | 4 |
| Q-053 | 0.97 | 3 |
| Q-056 | 2.17 | 117 |
| Q-057 | 0.99 | 1 |
| Q-058 | 1.10 | 1 |
| Q-060 | 2.86 | 186 |
| Q-061 | 0.51 | 49 |
| Q-065 | 1.32 | 32 |
| Q-068 | 1.08 | 8 |

Table C2 – Summary of the number of raingauges used to calculate reference rainfall, the number of combinations of these raingauges, and the most important raingauges for hydrological modeling.

| Catchment name | Catchment area [km²] | Catchment code | Number of reference raingauges | Number of combinations without missing data | Relevant raingauges |
|---|---|---|---|---|---|
| AMONT DU BASSIN - RIVIERE GRISE | 274.33 | Q-001 | 7 | 110 | P-091 P-104 P-095 P-118 |
| ARCAHAIE - RIVIERE MATHEUX | 65.65 | Q-004 | 7 | 92 | P-056 P-057 P-059 |
| BASSIN PROBY - RIVIERE COUJOL | 78.73 | Q-006 | 6 | 44 | P-054 P-056 P-057 P-059 P-087 P-108 |
| BUISSONNIERE - RIVIERE MOMANCE | 239.68 | Q-007 | 7 | 118 | P-091 P-114 P-118 |
| CAMP PERRIN - RIVIERE RAVINE DU SUD | 65.73 | Q-008 | 5 | 8 | P-116 P-135 P-136 P-150 P-131 |

| | | | | | |
|---|---|---|---|---|---|
| CAVAILLON - RIVIERE CAVAILLON | 321.21 | Q-010 | 6 | 39 | P-136 P-143 P-150 |
| GROS MORNE - RIVIERE TROIS-RIVIERES | 272.60 | Q-023 | 8 | 187 | P-001 P-004 P-033 P-068 P-070 |
| HINCHE - RIVIERE GUAYAMOUIC | 1966.90 | Q-024 | 11 | 1022 | P-075 P-017 P-056 P-059 P-068 P-069 P-070 P-054 P-033 |
| LES CAYES - RIVIERE ISLET | 100.77 | Q-029 | 5 | 19 | P-136 P-143 P-150 |
| MESSAYE - RIVIERE TORCELLE | 73.00 | Q-036 | 7 | 88 | P-054 P-087 P-093 P-100 |
| MIREBALAIS - RIVIERE ARTIBONITE | 7464.22 | Q-037 | 11 | 1634 | P-057 P-060 P-065 P-100 P-068 P-070 P-010 P-069 P-075 |
| P0NT-SONDE - RIVIERE ARTIBONITE | 8604.47 | Q-041 | 11 | 1648 | P-056 P-060 P-065 P-066 P-100 P-057 P-069 P-075 |
| PASSE FINE - RIVIERE LA THEME | 304.04 | Q-044 | 5 | 12 | P-056 P-057 P-093 P-100 P-108 |
| PASSE JOLY - RIVIERE D'ENNERY | 192.13 | Q-045 | 7 | 107 | P-004 P-068 P-028 P-044 P-045 P-033 |
| PLAISANCE - RIVIERE TROIS-RIVIERES | 44.84 | Q-051 | 6 | 43 | P-004 P-068 P-070 |
| PONT BENOIT - RIVIERE ESTERE | 137.90 | Q-052 | 8 | 185 | P-057 P-059 P-068 P-056 P-066 P-075 P-053 |
| PONT CHRISTOPHE - RIVIERE LIMBE | 245.46 | Q-053 | 8 | 187 | P-004 P-009 P-010 P-068 P-044 |
| RIVIERE GRANDE RIV. DU NORD | 547.75 | Q-056 | 10 | 508 | P-068 P-009 P-017 P-025 P-027 P-065 |
| PONT PETION - RIVIERE FER-A CHEVAL | 479.04 | Q-057 | 8 | 216 | P-062 P-072 P-104 |
| PONT TOUSSAINT - RIVIERE MONTROUIS | 168.20 | Q-058 | 10 | 950 | P-028 P-056 P-068 P-057 P-087 |
| OUANAMINTHE - RIVIERE MASSAGRE | 315.01 | Q-060 | 4 | 8 | P-057 P-068 P-017 |
| ROCHE HALEINE - RIVIERE LIMBE | 128.40 | Q-061 | 8 | 187 | P-004 P-033 P-044 P-068 P-070 P-010 P-009 |

| | | | | |
|---|---|---|---|---|
| ST-RAPHAEL - RIVIERE BONYAHA | 177.95 | Q-065 | 8 | 126 | P-004 P-009 P-045 P-068 P-010 P-033 |
| TORBECK - RIVIERE TORBECK | 95.99 | Q-068 | 4 | 9 | P-136 P-143 P-150 |

Table C3 - List of catchment attributes used in this study.

| Attribute class | Attribute name | Description | Unit | Data used |
|---|---|---|---|---|
| Location and topography | code | catchment identifier | - | The digital elevation model with a resolution of 90 m from SRTM (Reuter et al., 2007) and the catchment contours delineated in section 3.1.2 were utilized. |
| | name | Catchment name | - | |
| | Lon_Exu | Longitude of the catchment outlet | °W | |
| | Lat_Exu | Latitude of the catchment outlet | °N | |
| | Lon_Cent | Longitude of the catchment centroid | °W | |
| | Lat_Cent | Latitude of the catchment centroid | °N | |
| | Area | Catchment area | Km² | |
| | Gravelius | Gravelius coefficient (catchment elongation) | - | |
| | Min_Elev | Minimum catchment elevation | m | |
| | Max_elev | Maximum catchment elevation | m | |
| | Sd_Elev | Standard deviation of catchment elevations | m | |
| | Stream_density | ratio of the total of all stream segments to the area of the catchment | Km/km² | |
| | Slope | Average slope of the catchment computed according to Horn (1981) | ° | |

| | Hypso_curve | cumulative frequency of catchment elevations | m | |
|---|---|---|---|---|
| Geological characteristics | Lithology | Percentage of the catchment covered by each geologic class | % | The shapefile for CNIGS lithology classes was utilized. |
| | Carb_Rocks_Perc | Percentage of the catchment covered by carbonate sedimentary rocks | % | |
| | Sedim_Perc | Percentage of the catchment covered by sedimentary rocks | % | |
| | Magma_Perc | Percentage of the catchment covered by magmatic rocks | % | |
| Aquifer characteristic | Aquifer | Percentage of the aquifer classes by each catchment | % | The shapefile for CNIGS aquifer classes was utilized. |
| Land cover | Cover_95 | Percentage of the catchment covered by each landcover class (1995) | % | The shapefile for CNIGS land cover classes was utilized. |
| | Cover_98 | Percentage of the catchment covered by each landcover class (1998) | % | |
| Climatic indices | Aridity | Aridity index: ratio between the rainfall and the potential evapotranspiration (PET) | - | The rainfall, PET and temperature series at catchment scale described in section 3.2 were used. |
| | P_mean | Rainfall average | mm/month | |
| | T_mean | Temperature average | °C | |
| | PET_mean | PET average | mm/month | |
| | P_5_month | Rainfall quantile 5% | mm/month | |
| | T_5_month | Temperature quantile 5% | °C | |

| | PET_5_month | PET quantile 5% | mm/month | |
|---|---|---|---|---|
| | P_95_month | Rainfall quantile 95% | mm/month | |
| | T_95_month | Temperature quantile 95% | °C | |
| | PET_95_month | PET quantile 95% | mm/month | |
| | PMNA5 | Yearly minimum of monthly rainfall not exceeded once in 5 years | mm/month | |
| | PMXA10 | Yearly maximum of monthly rainfall exceeded once in 10 years | mm/month | |
| Hydrological signatures on a monthly time scale calculated with observed and 3 simulated streamflow | Runoff_Ratio | Runoff coefficient: ratio between the streamflow and the rainfall | - | The 24 streamflow series selected in section 3.1.1 and the parameters of the GR2M (Mouelhi et al., 2006) and GR4J (Perrin et al., 2003) rainfall-runoff models were used. |
| | Q_mean_month | Mean monthly streamflow | mm/month | |
| | Q_5_month | Streamflow quantile 5% | mm/month | |
| | Q_95_month | Streamflow quantile 95% | mm/month | |
| | QMNA5 | Yearly minimum of monthly streamflow not exceeded once in 5 years | mm/month | |
| | QMXA10 | Yearly maximum of monthly streamflow exceeded once in 10 years | mm/month | |
| | GR2M_param | the two parameters of GR2M | - | |
| Hydrological signatures on daily time scale calculated with observed and 3 simulated streamflow | Q_mean_day | Mean daily streamflow | mm/d | |
| | bfi | Baseflow index : ratio between the baseflow volume and the total streamflow volume | - | |

| | | (Pelletier and Andréassian, 2020) | |
|---|---|---|---|
| | high_q_freq | Frequency of high-flow days (> 9 times the median daily flow) | d/yr |
| | high_q_dur | Average duration of high-flow events (number of consecutive days > 9 times the median daily flow) | d |
| | low_q_freq | Frequency of low-flow days (< 0.2 times the mean daily flow) | d/yr |
| | low_q_dur | Average duration of low-flow events (number of consecutive days < 0.2 times the mean daily flow) | d |
| | Q_5_day | Streamflow quantile 5% (low flow) | mm/d |
| | Q_95_day | Streamflow quantile 5% (high flow) | mm/d |
| | GR4J_param | the four parameters of GR2M | |

Table C4 – Classes of lithology, aquifers, and land cover.

| lithology types | Alluvium, detrital materials |
|---|---|
| | Andesites and rhyodacites |
| | Basalt |
| | Diorite and tonalite |
| | Flysch, sandstone and limestone |
| | Hard limestone |
| | Marl and marly limestone |
| | Marl and sand |

| | Marly limestone |
|---|---|
| | Ultrabasic rocks |
| | Volcano-sedimentary rock |
| Acquifer types | Alluvial aquifers with free water |
| | Alluvial aquifers with partly confined water |
| | Carbonate aquifers with marl intercalation |
| | Crystalline formation |
| | Fissured and partitioned carbonate aquifers |
| | Highly permeable fissured and porous carbonate aquifers |
| | Karst aquifer |
| | Low permeability sedimentary formation |
| | More productive alluvial area |
| land use types in 1995 | Closed Shrubland |
| | Cropland |
| | Deciduous Broadleaf Forest |
| | Deciduous Needleleaf Forest |
| | Evergreen Broadleaf Forest |
| | Evergreen Needleleaf Forest |
| | Grassland |
| | Mixed Forest |
| | Open Shrubland |
| | Urban |
| | Water |
| | Wooded Grassland |
| | Woodland |
| land use types in 1998 | Beaches and dunes |

| | |
|---|---|
| | Continuous urban |
| | Dense agricultural crops |
| | Dense agroforestry systems |
| | Discontinuous urban |
| | Dominant pastures |
| | Forest |
| | Industrial areas |
| | Mangroves |
| | Medium-density agricultural crops |
| | Pasture with other presence |
| | Ports and airports |
| | Quarry |
| | River beds and recent alluvium |
| | Rock outcrops and bare soil |
| | Saline areas |
| | Savannahs |
| | Savannahs with other presence |
| | Water plan |
| | Wetlands |

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
