# Peer review of "Simbi: historical hydro-meteorological time series and signatures for 24 catchments in Haiti"

_Earth System Science Data, 2023_

## Referee Comment (RC1)

Review of

"Simbi: historical hydro-meteorological time series and signatures for 24 catchments in Haiti" by Bathelemy et al. submitted to Earth Systems Science Data

**1. Clarification**

I put my focus mainly on the structure of the dataset. It is therefore advisable to have at least two more reviews for the text part (introduction, methodology, discussion).

**2. General comment**

I appreciate the work that was done by the authors. Haiti is a data scarce region and every initiative to homogenize, proof and publish available data should be encouraged. But there are high uncertainties in the dataset. It must be considered that the quality of a dataset has a very large influence on the results of working groups, which are using it! The applied methodology is not novel, but this isn´t in my opinion a must in a data journal. The text is easy to understand. Summarizing information for the individual gauges with graphical summary sheets (e.g. Figure 15) is a nice way for giving a quick overview. Nevertheless, there are some points that should be improved before publication:

**3. Main comments**

a. The period of the available data is far back. What happened with the gauges afterwards? How were the measurements done such a long time ago (e.g. devices and it`s accuracy)? Information about those questions would give a deeper insight into the used data basis.
b. Is there a schedule for digitizing the more recent observed (runoff) data? Are all (or at least the most) gauges still in operation? If so, it would be very nice to do the work in the context of this paper and update your results. I think that more recent results are more relevant than those from the period 1920 to 1940.
c. The uncertainty of the results is in some parts high (KGE values < 0,5 are often referred to as poor results). I would add a chapter "Uncertainties" in or after the results to address and describe all sources of uncertainties (e.g. long time ago, measurement method, data aggregation, model). I am aware that uncertainties are omnipresent with such a data basis, but from my point of view it would be helpful to list them clearly and centrally at one place in your manuscript.
d. In addition, I would make a very clear statement at the end and also in your abstract about the high uncertainties. Otherwise, there is the risk that further working groups do not sufficiently consider the high uncertainties.
e. The used data sources (e.g. land cover) and method for calculation the catchment attributes are not always clearly specified and listed. This is a lack of transparency.
f. Try to specify the data products used for calculating your catchment attributes (e.g. year of publication, domain, spatial resolution, update frequency, reference) at least in a list in the appendix.
g. The structure of your dataset isn´t friendly for machine-reading. Details are listed below.
h. Sharing the codes is helpful for reproducing your results and would make it easier for other groups to build upon your work.
i. Links should be listed in the references and not in the text, in order not to disrupt the reading flow.

**4. Minor comments**

They are just a first glimpse for the start, see clarification.

a. L45: The CAMELS datasets also provide indices and signatures.

b. L111: Can the digitization process be described (e.g. with an indication of accuracy)?

c. L177: Is the final error declared with an attribute?

d. L181: What Is indicated by the white, orange and blue dots? A legend would make it much easier to understand the figure.

e. L238: Why were the two individual datasets NOAA 20CR and BEST area-weighted averaged? Can a dataset be used without the other one?

f. L273: How was the aggregation done (e.g. area-weighting of the intersecting pixel or weighted all pixels equal not matter of the intersected individual area) and what was the output (e.g. mean, median). This information would be important.

g. L276: What is relevant to Haitian catchments and what not? I think the most readers do not have this information.

h. L279: Did you use the R-codes of Nans Addor (CAMELS-US) to calculate the hydrological signatures? If yes, it would be good to mention that.

i. L287: It seems that the two datasets for land cover are different data sources (other attributes). Can you give details about the differences (e.g. data genesis).

j. L288: There are global land use datasets (mostly derived by earth observation methods) for the most recent period.

k. L293: What is meant with "cut out"?

l. L367: What is the red line and the red point indicating in the figure?

m. L376: Try to be more precise, what is a critical percentage of missing data?

n. L392: If the relevant raingauge combination can cause such an error, have we to conclude that this amount of deviation is also possible at the other catchments? By the way: Can you declare the weights of the applied raingauge combination in a text file?

o. L423: There is partly a big difference of mean annual discharge between neighbouring gauges (> 600 mm/year). Can you explain that?

p. L470: This sentence fits better in the conclusion.

q. L484: How where the return periods calculated? Which statistical distribution?

r. L492: I would set the chapter Data availability after Conclusion.

s. L518: There are also outliers and not really a clear trend, see Figure 9.

t. L600: An additional column with the source dataset would enhance transparency in Table C1, C2 and C3.

**5. Dataset**

a. It is not user-friendly, that your dataset is compressed 2-fold. 1-fold (all files together) is enough.

b. I would name the compressed file Simbi and not "dataverse_files".

c. The dataset is unnecessarily nested after extracting (e.g. dataverse_files\SIMBI\SIMBI_00_OBSERVED_DATA\00_OBSERVED_DATA\). Folder levels could be removed.

d. Storing the files as .csv instead .txt would enable an easy and fast opening with the software Excel. This is nice for a quick view.

e. The files are not friendly for machine-reading. I would remove the description in the header of your data files and create a new file with the belonging metadata. In this metadata file it is recommended to list the important information (e.g. Station-code, unit) directly after the name of the variable without blanks or special characters, e.g.:
Station_code;P-001
Unit;mm/d
→ Hint: Get rid of the blanks in the metadata, they are good for human reading, but not for machine reading.

f. It is tedious to read and separate lines like line 10 in Q_001___AMONT_DU_BASSIN___RIVIERE_GRISE (## Station code        : 40501    ; Q-001 ; (DHSI        ; URGEO). Try to get the DHSI station code (40501) with R, then you see why.

In general, try to make it as easy as possible to import data and the belonging metadata e.g. with the software R.

g. I would rename the files with the timeseries (e.g. instead Q_001___AMONT_DU_BASSIN___RIVIERE_GRISE → Q_1)

h. I would indicate the gaps in the time series not with a character (NA), simply do not enter anything. This ensures the data format numeric.

i. The ID of the gauge (code) should always be in the first column (this is not the case e.g. in the file location_and_topography.txt).

j. The number of decimal places is sometimes really too high (e.g. Percent_geologic_class.txt). This implies an accuracy which is not given.

k. The stream density includes a lot of information of a catchment (e.g. hint for carstic region). I would calculate and add this important catchment attribute. If there is no local dataset with the Haitian streams, take a global dataset like HydroATLAS (Linke et al., 2019, https://doi.org/10.1038/s41597-019-0300-6)

l. Have a look at Gudmundsson et al. (2018, https://doi.org/10.5194/essd-10-787-2018) regarding quality control flags (e.g. value does not change over more than x timesteps, outlier) for your timeseries.

m. Adding metadata to your shapefiles would be helpful, where the header and the unit of your attributes is declared.

**6. Plots**

a. The colour scales of your plots are not always intuitive (e.g. Figure 4 or 9). There is a nice article from Stoelzle and Stein (2021, https://doi.org/10.5194/hess-25-4549-2021) about visualization in hydrology. I recommend considering this guidance for your figures.

b. The font sizes in your plots vary sometimes greatly (e.g. Figure 7). It would be good to homogenize the sizes.

c. It is often more intuitive for the reader to extend the legend of a plot, than describing the attributes in the label of the figure (e.g. Figure 1).

---

## Referee Comment (RC4)

Haiti is acutely vulnerable to natural disasters, yet it lacks an accessible hydroclimatic database for analysis. The paper provides a meaningful database, known as Simbi, to fill this gap. The database is undoubtedly valuable, and the paper is well-written. I only have minor comments as follows:

1. it would be beneficial if the authors include a figure or a table detailing the data types, temporal extent, and other related information.

2. In the Simbi database, some data have been sourced from previous studies. It is essential to explicitly describe the origins of the data within the database.

3. In section 3.1.1, a rationale is needed to provide for selecting streamflow series that are hydrologically relevant.

4. In Section 3.2.1, the description of "multiple raingauge combinations" lacks clarity. If the number of rain gauges is fewer than three, it is unclear how to utilize Thiessen polygons.

5. The KGE equation is not correct. The equation in the bracket should be $(1-r)^2+(1-\alpha)^2+(1-\beta)^2$.

6. In Figure 10, the average monthly streamflows of 48 catchments were calculated. However, there is considerable uncertainty due to data discrepancies in many catchments. Therefore, it is recommended to include confidence intervals in Figure 10.

7. In Figure 10, providing reasons for the observed lags in the trend between rainfall and streamflow in Nov.

8. According to Simbi, can you provide a summary of the hydroclimatic characteristics of this region?

---

## Author Comment (AC1)

**Simbi: historical hydro-meteorological time series and signatures for 24 catchments in Haiti**

**Manuscript No. ESSD-2023-259**

Ralph Bathelemy, Pierre Brigode, Vazken Andréassian, Charles Perrin, Vincent Moron, Cédric Gaucherel, Emmanuel Tric, and Dominique Boisson.

**Reply to reviewer #1**

We thank reviewer #1 for this detailed review and helpful comments. Please find below our replies to the reviewer's comments. We provided specific responses (in black) to the reviewer comments (in *italic and blue*).

1. **General comment**

   *I appreciate the work that was done by the authors. Haiti is a data scarce region and every initiative to homogenize, proof and publish available data should be encouraged. But there are high uncertainties in the dataset. It must be considered that the quality of a dataset has a very large influence on the results of working groups, which are using it! The applied methodology is not novel, but this isn´t in my opinion a must in a data journal. The text is easy to understand. Summarizing information for the individual gauges with graphical summary sheets (e.g. Figure 15) is a nice way for giving a quick overview. Nevertheless, there are some points that should be improved before publication:*

   We thank the reviewer #1 for this positive feedback.

2. **Main comments**

   a. *The period of the available data is far back. What happened with the gauges afterwards? How were the measurements done such a long time ago (e.g. devices and it`s accuracy)? Information about those questions would give a deeper insight into the used data basis.*

   Streamflow measurements on Haitian rivers were conducted during the American occupation (1915-1934) under the supervision of USGS engineers. After the end of the American occupation in 1934, the number of active stations gradually decreased until 1940 when streamflow measurements ceased. Although some measurements were resumed in the 1960s and again in the 1980s, most of the data is now lost, and what remains is very fragmentary and only available for short periods. Consequently, the only reliable streamflow measurements are obtainable for the period 1920-1940.

   Graduated vertical rulers were installed on one riverbank and read two to three times daily. On average, each station collected 12 gauging measurements annually. Between 1925 and 1928, 11 of the 70 stations replaced the vertical rulers with automatic recorders; however, no information was available on the type of equipment installed. Hydrographic bulletins also reported on the quality of the rating curves (fair rating curve, very good rating curve for medium flows, fair rating curve for high flows, etc.).

   This information will be provided in the next version of the manuscript. Additionally, metadata will include information about the instrumentation, whether it was manual or automatic, and the rating curve quality for each station.

*b. Is there a schedule for digitizing the more recent observed (runoff) data? Are all (or at least the most) gauges still in operation? If so, it would be very nice to do the work in the context of this paper and update your results. I think that more recent results are more relevant than those from the period 1920 to 1940.*

After 1940, there is a scarcity of streamflow observation data available in Haiti. Good quality streamflow data can only be found for the period between 1920-1940, which has been presented in Simbi.

We will describe the evolution of hydrometry in Haiti since 1940 in the next version of the manuscript.

*c. The uncertainty of the results is in some parts high (KGE values < 0,5 are often referred to as poor results). I would add a chapter "Uncertainties" in or after the results to address and describe all sources of uncertainties (e.g. long time ago, measurement method, data aggregation, model). I am aware that uncertainties are omnipresent with such a data basis, but from my point of view it would be helpful to list them clearly and centrally at one place in your manuscript.*

Thanks for this suggestion: we will add a dedicated section devoted to the database uncertainties in the next version of the manuscript, to give comments on the various sources of uncertainties and their potential impacts. This should be helpful for the users of the datasets.

*d. In addition, I would make a very clear statement at the end and also in your abstract about the high uncertainties. Otherwise, there is the risk that further working groups do not sufficiently consider the high uncertainties.*

Uncertainties will be clearly stated in the conclusion of the manuscript and in the abstract in the next version of the manuscript, to warn working groups of the possible pitfalls associated with the use of this dataset.

*e. The used data sources (e.g. land cover) and method for calculation the catchment attributes are not always clearly specified and listed. This is a lack of transparency.*

The next version of the manuscript will provide a more detailed description of the data products used, including land cover, geological, and aquifer type data. Additionally, Table C2 will include the mathematical formulations and references for each attribute used.

*f. Try to specify the data products used for calculating your catchment attributes (e.g. year of publication, domain, spatial resolution, update frequency, reference) at least in a list in the appendix.*

The catchment-scale hydro-climatic data produced in this article and the land-use, geological, aquifer type and topographical data were used to calculate the attributes.

The next version of the manuscript will list the information of the data products used to calculate catchment attributes (e.g. year of publication, domain, spatial resolution, update frequency, and reference).

*g. The structure of your dataset isn´t friendly for machine-reading. Details are listed below.*

The structure of the dataset has been modified for machine-reading (see the dataset section below).

*h. Sharing the codes is helpful for reproducing your results and would make it easier for other groups to build upon your work.*

R codes reading Simbi on all catchments and performing rainfall-runoff model calibration and simulation over the entire period will be provided with the next version of the manuscript.

i. *Links should be listed in the references and not in the text, in order not to disrupt the reading flow.*

The links will be removed from the text, except for the link in the abstract because the dataset link must be included in the abstract according to the ESSD manuscript preparation guidelines.

3. **Minor comments**

*They are just a first glimpse for the start, see clarification.*

a. *L45: The CAMELS datasets also provide indices and signatures.*

The sentence will be replaced by: *"While the CAMELS databases provide time series, indices and hydroclimatic signatures of catchments, other databases provide only indices and hydroclimatic signatures of catchments, such as the African Database of Hydrometric Indices (ADHI; Tramblay et al., 2021)".*

b. *L111: Can the digitization process be described (e.g. with an indication of accuracy)?*

The digitization process is described in Appendix A. However, it will be described in greater detail in the next version of the manuscript.

c. *L177: Is the final error declared with an attribute?*

The final error will be declared with an attribute in the next version of the manuscript.

d. *L181: What Is indicated by the white, orange and blue dots? A legend would make it much easier to understand the figure.*

The white, orange and blue dots indicated all raingauges with monthly data for the period 1920–1940. Raingauge stations with air temperature data are shown in orange. Raingauges considered relevant for hydrological modeling are shown in blue (see section **Erreur ! Source du renvoi introuvable.**). Other raingauges are shown in white. A legend will be added in the next version of the manuscript.

e. *L238: Why were the two individual datasets NOAA 20CR and BEST area-weighted averaged? Can a dataset be used without the other one?*

Both air temperature databases were used independently. The wording of the previous sentence was unclear, so it will be revised in the upcoming version of the manuscript:

*Catchment air temperature series were computed on a daily time step for two temperature databases (NOAA 20CR and BEST) by taking the weighted average of pixels in the respective database (NOAA 20CR or BEST).*

f. *L273: How was the aggregation done (e.g. area-weighting of the intersecting pixel or weighted all pixels equal not matter of the intersected individual area) and what was the output (e.g. mean, median). This information would be important.*

Catchment scale data are computed as a weighted average. The weights are proportional to the area of the pixel overlapping the catchment. This chapter will be described in greater detail in the next version of the manuscript.

g. *L276: What is relevant to Haitian catchments and what not? I think the most readers do not have this information.*

Thanks for this suggestion. This sentence will be reworded:

*A set of attributes that describes a broad range of low, moderate, and extreme precipitation and streamflow characteristics were chosen to characterize the hydrological regime, based on available data.*

h. *L279: Did you use the R-codes of Nans Addor (CAMELS-US) to calculate the hydrological signatures? If yes, it would be good to mention that.*

We did not use the R codes of Nans Addor to calculate hydrological signatures.

i. *L287: It seems that the two datasets for land cover are different data sources (other attributes). Can you give details about the differences (e.g. data genesis).*

Thanks for this suggestion. Details of the differences between the two land cover datasets will be provided in the next version of the manuscript.

j. *L288: There are global land use datasets (mostly derived by earth observation methods) for the most recent period.*

The existence of these alternative datasets will be mentioned in the next version of the manuscript.

k. *L293: What is meant with "cut out"?*

The expression "cut out" will replaced by "cropped".

l. *L367: What is the red line and the red point indicating in the figure?*

The red line represents the optimal ratio (r=1), while the red dot represents the mean value of the distribution. The figure caption will be modified:

*Ratio of the GR2M calibrated parameters X1 and X2 over the two subperiods for the reference and relevant raingauge combinations. The red line represents the optimal ratio (r=1), while the red dot represents the mean value of the distribution.*

m. *L376: Try to be more precise, what is a critical percentage of missing data?*

If a station has less than 10 years of data, it will be deemed to have a critical percentage of missing data (The 10-year threshold was set arbitrarily). This paragraph will give more detail in the next version of the manuscript.

n. *L392: If the relevant raingauge combination can cause such an error, have we to conclude that this amount of deviation is also possible at the other catchments? By the way: Can you declare the weights of the applied raingauge combination in a text file?*

If the relevant rain gauges are wet or dry, a greater or lesser deviation may be observed. The weights of the rain gauge combination applied will be presented in a dedicated file.

o. *L423: There is partly a big difference of mean annual discharge between neighbouring gauges (> 600 mm/year). Can you explain that?*

The Q-008 watershed has a significantly higher mean annual streamflow than the three neighboring watersheds (Q-010, Q-068 and Q-029). As indicated in line 387, over 90% of the Q-008 watershed is located on a limestone geological formation, and there is also an impact from karst aquifers within the watershed. On this basis, it is likely that an influx of water from neighboring catchments is responsible for such a high mean annual streamflow. However, no studies have been carried out to verify or challenge this hypothesis. This point will be highlighted in the next manuscript version.

p. *L470: This sentence fits better in the conclusion.*

Thanks for this suggestion. This sentence will be moved in the conclusion.

q. *L484: How where the return periods calculated? Which statistical distribution?*

The Generalized Extreme Value distribution and the distribution of annual values (precipitation, PE, air temperature or streamflow) were used to estimate values for multiple return periods. This will be clarified in the next manuscript version:

*Simulated streamflows underestimate maximum annual flows with a return period of less than 10 years and overestimate flows beyond 10 years. The generalized extreme value (Beirlant et al., 2004; Coles, 2001; Jenkinson, 1955) and the distribution of annual*

*values (precipitation, PE, air temperature or streamflow) were used to estimate values for multiple return periods.*

r.  *L492: I would set the chapter Data availability after Conclusion.*

The data availability section must be placed before the conclusion section according to the ESSD manuscript preparation guidelines.

s.  *L518: There are also outliers and not really a clear trend, see Figure 9.*

Thanks for this comment. Indeed, the central part of Haiti is associated with low streamflow values and the southwest with high streamflow values. However, no clear trend was observed in the north. This analysis will be changed in the next manuscript version:

*The central part of Haiti is associated with relatively low streamflow and high drought coefficients. The southwest is associated with relatively high streamflow. In fact, large floods are more frequent in these areas (Terrier et al., 2017). No clear trend was observed in the north.*

t.  *L600: An additional column with the source dataset would enhance transparency in Table C1, C2 and C3.*

An additional column with the source dataset will be added in the next version of the manuscript.

4.  **Dataset**

a.  *It is not user-friendly, that your dataset is compressed 2-fold. 1-fold (all files together) is enough.*

Thank you for your comment. Unfortunately, this two-time compression is due to the data warehouse that adds extra folders, and we are not able to change that.

b.  *I would name the compressed file Simbi and not "dataverse_files".*

The folder "dataverse_files" is added by the data warehouse and is not reliant on us.

c.  *The dataset is unnecessarily nested after extracting (e.g. dataverse_files\SIMBI\SIMBI_00_OBSERVED_DATA\00_OBSERVED_DATA\). Folder levels could be removed.*

The first three levels (dataverse_files\SIMBI\SIMBI_00_OBSERVED_DATA) are automatically included by the data warehouse and therefore cannot be removed. These directories are intended for use by other databases also hosted within this warehouse, such as the ADHI database (https://doi.org/10.23708/LXGXQ9).

d.  *Storing the files as .csv instead .txt would enable an easy and fast opening with the software Excel. This is nice for a quick view.*

The files are now saved in csv format.

e.  *The files are not friendly for machine-reading. I would remove the description in the header of your data files and create a new file with the belonging metadata. In this metadata file it is recommended to list the important information (e.g. Station-code, unit) directly after the name of the variable without blanks or special characters, e.g.: Station_code;P-001 Unit;mm/d → Hint: Get rid of the blanks in the metadata, they are good for human reading, but not for machine reading.*

The headers of the data files have been deleted and new files have been created to store the metadata. The metadata files include:

i.   a file containing information on rain gauge stations,

ii.  a file containing information on streamflow stations,

iii. a file containing monthly time-step modeling information, and

iv.  a file containing daily time-step modeling information.

f. *It is tedious to read and separate lines like line 10 in Q_001___AMONT_DU_BASSIN___RIVIERE_GRISE (## Station code : 40501 ; Q-001 ; (DHSI ; URGEO). Try to get the DHSI station code (40501) with R, then you see why. In general, try to make it as easy as possible to import data and the belonging metadata e.g. with the software R.*

Metadata is now stored in separate files in a tabular format, allowing for easy importation.

g. *I would rename the files with the timeseries (e.g. instead Q_001___AMONT_DU_BASSIN___RIVIERE_GRISE → Q_1)*

Files have been renamed using station codes to facilitate identification (e.g. Q_001 instead of Q_001___AMONT_DU_BASSIN___RIVIERE_GRISE).

h. *I would indicate the gaps in the time series not with a character (NA), simply do not enter anything. This ensures the data format numeric.*

In certain files, one column contains observed streamflows (including gaps) while another column contains simulated streamflows (without gaps). This discrepancy complicates the removal of rows with missing observed data. To maintain numeric consistency, the NA character is now substituted with the value of -9999.

i. *The ID of the gauge (code) should always be in the first column (this is not the case e.g. in the file location_and_topography.txt).*

Thanks for this comment. Station IDs are now in the first column.

j. *The number of decimal places is sometimes really too high (e.g. Percent_geologic_class.txt). This implies an accuracy which is not given.*

Thanks for this comment. The decimal places are now limited to 2.

k. *The stream density includes a lot of information of a catchment (e.g. hint for carstic region). I would calculate and add this important catchment attribute. If there is no local dataset with the Haitian streams, take a global dataset like HydroATLAS (Linke et al., 2019, https://doi.org/10.1038/s41597-019-0300-6)*

Thanks for this suggestion. Stream density will be included as an attribute in the next version of the manuscript by utilizing dataset from CNIGS (National Center for Geospatial Information in Haiti).

l. *Have a look at Gudmundsson et al. (2018, https://doi.org/10.5194/essd-10-787-2018) regarding quality control flags (e.g. value does not change over more than x timesteps, outlier) for your timeseries.*

Thanks for this suggestion. In addition to the criteria outlined in section 3.1.1 for selecting streamflow series, the three indicators proposed by Gudmundsson *et al.* (2018) for quality control of time series have been added in Simbi:

   i. days for which Q<0, where Q denotes a daily streamflow value,

   ii. daily values with more than 10 consecutive equal values larger than zero, and

   iii. outlier detection.

m. *Adding metadata to your shapefiles would be helpful, where the header and the unit of your attributes is declared.*

Metadata (name, code, altitude, and area) and their units will be added to the shapefiles.

5. **Plots**

   a. *The colour scales of your plots are not always intuitive (e.g. Figure 4 or 9). There is a nice article from Stoelzle and Stein (2021, https://doi.org/10.5194/hess-25-4549-*

*2021) about visualization in hydrology. I recommend considering this guidance for your figures.*

Thank you for this comment. The next version of the manuscript will include modified color scales for the figures.

***b.*** *The font sizes in your plots vary sometimes greatly (e.g. Figure 7). It would be good to homogenize the sizes.*

The font sizes of the plots will be homogenized in the next manuscript version.

***c.*** *It is often more intuitive for the reader to extend the legend of a plot, than describing the attributes in the label of the figure (e.g. Figure 1).*

Figure attributes will be described in the figure legend in the next manuscript version.

---

## Author Comment (AC2)

**Simbi: historical hydro-meteorological time series and signatures for 24 catchments in Haiti**

**Manuscript No. ESSD-2023-259**

Ralph Bathelemy, Pierre Brigode, Vazken Andréassian, Charles Perrin, Vincent Moron, Cédric Gaucherel, Emmanuel Tric, and Dominique Boisson.

**Reply to reviewer #2**

We thank reviewer #2 for this detailed review and helpful comments. Please find below our replies to the reviewer's comments. We provided specific responses (in black) to the reviewer comments (in *italic and blue*).

1. **ABSTRACT**

    a. *Besides being "exposed to hydroclimatic hazards" is there anything else that would make the hydrology of the area interesting? The paper could clarify this point starting from the abstract.*

    It is a region with steep slopes that generate flash floods, especially in small catchments. Additionally, the hydrology of this region is poorly understood and little studied. Clarifications will be made in the next version of the manuscript.

    b. *What are the limitations of the dataset? Why is it that mainly the 1920-1940 period is analyzed?*

    Reviewer #1 recommends dedicating a section to database uncertainties, which will be summarized in the abstract. Study limitations and uncertainties will be described in the next version of the manuscript.

    The 1920-1940 period was chosen as it is the only time period with high-quality streamflow measurements available.

2. **INTRODUCTION**

    a. *P1L45 – CAMELS also provides hydroclimatic signatures?*

    The sentence will be replaced by: *"While the CAMELS databases provide time series, indices and hydroclimatic signatures of catchments, other databases provide only indices and hydroclimatic signatures of catchments, such as the African Database of Hydrometric Indices* (ADHI; Tramblay *et al.*, 2021)*".*

    b. *P2L47 – Define "these works".*

    "These works" is replaced by "this paper".

    c. *P2L50 – It reads like there is nothing for Haiti, but it seems that the dataset was built on previous data. Shouldn't the background be provided here?*

    Thanks for the comment. This article relies on data from the BVH project conducted in Haiti in 2012. This project and the associated dataset will be better described in the next version of the manuscript.

    d. *P2L53 – What is the definition of "anarchic urbanization"?*

    Anarchic urbanization is the practice of urbanizing in a way that disregards town planning regulations. Anarchic urbanization will be better described and illustrated in the next version of the manuscript.

e. *P2L63 – "These two databases…" See previous comment on line 50.*

The two databases are (i) the monthly rainfall database (1905-2005) built by Moron *et al.* (2015) and (ii) the 70 daily streamflow series (1920-1940) from the BVH project. The next version of the manuscript will provide more detailed descriptions of the BHV project and these two databases.

f. *P2L75 – Is "producing" the appropriate verb here?*

"Producing" will be replaced by "building".

g. *P2L77 – Why for the 1920-1940 period?*

The of 1920-1940 period is chosen as it is the only time period with high-quality streamflow measurements available.

3. **DATA USED**

a. *L90 – What is the meaning of 70 daily series, but 24 are used? Why 1920-1940? Provide a better description of missing data etc.*

70 streamflow series are available, with 24 being then selected for rainfall-runoff modeling based on their quality. The process for selecting streamflow is explained in section 3.1.1.

This section will be further detailed in the next manuscript version, incorporating comments from reviewer #1 and reviewer #2.

b. *L106 – Can you also provide a graph showing missing data periods and percentage?*

The percentages of missing data will be quantified and presented graphically in the next version of the manuscript.

c. *L122 – Exactly what significant gaps?*

An arbitrary threshold of five years has been set, beyond which missing data is considered significant. The percentages of missing data will be quantified and presented graphically in the next version of the manuscript.

4. **METHODOLOGY**

a. *L138 – Why for 21 of 24?*

24 streamflow series were selected from a total of 70. Monthly rainfall-runoff modeling was performed on those 24 associated catchments. However, only 21 of the 24 catchments had available daily rainfall series. Consequently, daily rainfall-runoff modeling was performed for these 21 catchments. This will be clarified in the next version of the manuscript.

b. *L159 – Why were the remainder not provided?*

These streamflow series were not selected because they are located downstream of diversion canals or small dams used for irrigation. They poorly represent streamflow seasonality and are therefore considered to be influenced by human activities. This paragraph will be reworded in the next article version.

c. *L169 – What is the meaning of "numerically calculated"?*

Catchment contours were delineated using algorithms (digitally) as opposed to contours estimated using topographic maps in historical times. This paragraph will be reworded in the next version of the manuscript.

d. *L171-172 – What does it mean to "relocate manually"?*

The provided geographical locations for certain stations did not correspond to the information in the hydrographic bulletins. The locations of these stations were

rectified utilizing the information available in the hydrographic bulletins. This paragraph will be reworded in the next version of the manuscript.

e. *L174 – What "numerical model"?*

The "numerical model" is the DEM (digital elevation model). This sentence will be reworded in the next version of the manuscript.

f. *L177 – This is not clear.*

Rivers generated with SRTM DEM differ from real rivers. Thus, station positions must be relocated to match the rivers generated by the DEM processing. This paragraph describes the station relocation process.

This paragraph will be better described in the next version of the manuscript.

g. *FIG.1 – Provide Haiti country boundary. Are the gray lines the river network? How to consider if a raingauge is relevant? What are the white dots?*

Gray lines indicate the river network; white dots represent non relevant raingauges. Relevant rain gauges are defined in section 3.2.1.

In the next version of the manuscript, a precise illustration of Haiti's boundaries and a more comprehensive figure legend will be provided.

h. *L187 – Is this the best method to identify appropriate raingauge?*

This method seems to be the most appropriate for identifying the relevant raingauges for rainfall-runoff modelling. Indeed, the performance of a rainfall-runoff model improves with a better description of the rainfall input (Andréassian *et al.,* 2001). The GR2M model was therefore used to determine the relevant raingauges in this study. Relevant raingauges are defined as those with the best performance of the model. The results with GR2M are a first estimate of relevant raingauges to ease future work.

The limit of this method is the dependence of the relevant raingauges on the GR2M model used. This paragraph will be better described in the next version of the manuscript.

i. *L213-214 – Can this be shown in a figure? What is a continuous catchment rainfall?*

Continuous catchment rainfall is a catchment-scale rainfall series with no missing data. This will be clarified in the next version of the manuscript.

j. *L216-221 – This is not clear.*

This paragraph describes the procedure for calibrating and evaluating the performance of the GR2M rainfall-runoff model over two distinct sub-periods, P1 and P2, for each catchment.

This paragraph will be better described in the next version of the manuscript.

k. *L222-224 – Why was it used then? This is contradictory to the previous explanation.*

This paragraph presents the limitations of the method used and explains the dependence of the results on the rainfall-runoff model used.

This paragraph will be better described in the next version of the manuscript.

l. *L253 – What is the meaning of "most relevant method"?*

This sentence has been reworded as follows: *"Moreover, it is one of the most relevant approach to calculate PET for use in rainfall–runoff compared to 27 models for calculating PET and has been tested on more than 300 catchments covering several climatic zones, including tropical zones (Oudin et al., 2005), as is the case in Haiti".*

m. *L263 – Parameters are called "Period 1"… this is not clear and not the best choice of terms.*

The three periods used for calibration will be referred to as P1, P2 and P3 in the next version of the manuscript.

n.  *L277 – SRTM and catchment attributes should be all explained in the same section.*

SRTM and topographic attributes will be explained in the same section.

o.  *L283 – to produce… are produced.*

This sentence has been reworded as follows: *"The data used to calculate the geological attributes, land cover characteristics, and aquifer types are produced by the CNIGS (Centre National de l'Information Géospatiale) and the BME (Bureau des Mines et de l'Energie)".*

p.  *L288-292 – Should be clearly explained/justified in the beginning. I figure showing the changes would be interesting.*

This paragraph will be better presented in the next version of the manuscript, taking into account your comments and those of reviewer #1. Geology, aquifer types and land use datasets will be better described. Additionally, we will provide illustrations showcasing the evolution of land use.

q.  *Fig. 3, 5, 6, and 8 should be moved to supplement. This is not well explained and get in the way of the analysis.*

Figures 3, 5, 6 and 8 will be moved to the supplement in the next version of the manuscript.

r.  *Fig 4 and 9 – The color pallet is not appropriate.*

An appropriate color pallet will be used in the next version pf the manuscript.

s.  *Fig 10 – Why this figure? Is this an average for the entire country? How relevant for understanding the hydrology of the region is this?*

This figure shows rainfall and streamflow (observed and simulated) regimes. It shows i) the seasonality of rainfall and streamflow in Haiti, with two periods of heavy rainfall/streamflow around May and between September and November, and ii) a comparison of observed and simulated streamflow regimes. Each of the three regimes is calculated using data averaged over the entire country.

The paragraph describing this figure will be better described in the next version of the manuscript.

t.  *Fig 13 and 14 – Instead of this bars, isn't it better to provide a map of the geology and aquifer type of the region? Check most of the CAMELS papers for good examples.*

Thanks for your comment. We will draw inspiration from the CAMELS databases to better represent geology and aquifer types.

u.  *Fig 15 – This is a great example of useful graphics for having a quick look at the data and also precious information for decision makers.*

Thanks for this comment.

5.  **CONCLUSION AND PERSPECTIVE**

a.  *L528 – "over several decades" seems to be a stretch when considering 1920-1940.*

"Over several decades" will be replaced by "over several years".

b.  *L531 – Please, define and show in a figure what is meant by massive deforestation and anarchic urbanization, otherwise, delete it.*

The massive deforestation and the anarchic urbanization will be defined and illustrated with figures in the next version of the manuscript.

c.  *L533-535 – Strange choice of words for these two sentences. Maybe delete?*

These two sentences have been reworded as follows:

*"Frequency analysis methods can be utilized to estimate flood return periods. The accessibility of streamflow data allows for the possibility of various rainfall-runoff modeling approaches to be applied".*

**6. REFERENCES**

Andréassian, V., Perrin, C., Michel, C., Usart-Sanchez, I., Lavabre, J., 2001. Impact of imperfect rainfall knowledge on the efficiency and the parameters of watershed models. Journal of Hydrology 250, 206–223. https://doi.org/10.1016/S0022-1694(01)00437-1

Moron, V., Frelat, R., Jean-Jeune, P.K., Gaucherel, C., 2015. Interannual and intra-annual variability of rainfall in Haiti (1905–2005). Clim Dyn 45, 915–932. https://doi.org/10.1007/s00382-014-2326-y

Tramblay, Y., Rouché, N., Paturel, J.-E., Mahé, G., Boyer, J.-F., Amoussou, E., Bodian, A., Dacosta, H., Dakhlaoui, H., Dezetter, A., Hughes, D., Hanich, L., Peugeot, C., Tshimanga, R., Lachassagne, P., 2021. ADHI: the African Database of Hydrometric Indices (1950–2018). Earth System Science Data 13, 1547–1560. https://doi.org/10.5194/essd-13-1547-2021

---

## Author Response (AR1)

**Simbi: historical hydro-meteorological time series and signatures for 24 catchments in Haiti**

**Manuscript No. ESSD-2023-259**

Ralph Bathelemy, Pierre Brigode, Vazken Andréassian, Charles Perrin, Vincent Moron, Cédric Gaucherel, Emmanuel Tric, and Dominique Boisson.

**Point-by-point reply to the reviewers**

We thank the Reviewers for their detailed reviews and helpful comments. Please find below our replies to the Reviewers' comments. We provided the reviewer comments *in italic and blue*, the author's responses in black and the author's changes in the manuscript in *black and italic*.

**A – ANSWER TO THE REVIEWER #1**

1. **General comment**

    *I appreciate the work that was done by the authors. Haiti is a data scarce region and every initiative to homogenize, proof and publish available data should be encouraged. But there are high uncertainties in the dataset. It must be considered that the quality of a dataset has a very large influence on the results of working groups, which are using it! The applied methodology is not novel, but this isn´t in my opinion a must in a data journal. The text is easy to understand. Summarizing information for the individual gauges with graphical summary sheets (e.g. Figure 15) is a nice way for giving a quick overview. Nevertheless, there are some points that should be improved before publication:*

2. **Main comments**

    a. *The period of the available data is far back. What happened with the gauges afterwards? How were the measurements done such a long time ago (e.g. devices and it`s accuracy)? Information about those questions would give a deeper insight into the used data basis.*

    Streamflow measurements on Haitian rivers were conducted during the American occupation (1915-1934) under the supervision of USGS engineers. After the American occupation in 1934, the number of active stations gradually decreased until 1940 when streamflow measurements ceased. Although some measurements were resumed in the 1960s and again in the 1980s, most of the data is lost, and what remains is very fragmentary and only available for short periods. Consequently, the only reliable streamflow measurements are obtainable for the period 1920-1940.

    Graduated vertical rulers were installed on one riverbank and read two to three times daily. On average, each station collected 12 gauging measurements annually. Between 1925 and 1928, 11 of the 70 stations replaced the vertical rulers with automatic recorders; however, no information was provided on the type of equipment installed. Hydrographic bulletins also reported on the quality of the rating curves (fair rating curve, very good rating curve for medium flows, fair rating curve for high flows, etc.).

    these two paragraphs in the introduction have been revised. Additionally, metadata files have been added to the database and included information about the instrumentation, whether it was manual or automatic, and the calibration curve quality for each station. The changes in the manuscript are:

    *Unfortunately, there are significant differences between countries in terms of the quality and quantity of hydroclimatic reference databases, as well as regarding access*

*to these data. Some countries do not have such reference databases. This is the case of Haiti, whose territory is, moreover, highly exposed to natural disasters (Khouakhi et al., 2017; Burgess et al., 2018), and climate change (Peterson et al., 2002). At the same time, Haiti is facing the consequences of massive deforestation and anarchic urbanization (urban development that does not comply with planning regulations) in recent decades (Hedges et al., 2018; Tarter et al., 2018; Mompremier et al., 2022), resulting in increased vulnerability to hydroclimatic hazards. Currently, Haiti lacks a freely and easily accessible hydroclimatic database due to the absence of in situ hydroclimatic observations. The first hydrometric observations were conducted during the American occupation of Haiti, and began in 1919. American engineers from the Water Resources Service (WRS) of the United States Geological Survey (USGS) supervised these hydrological observations, that continued into the 1940s and exceptionally later. The end of the American occupation is the main reason for the cessation of hydrometric observations. This is due to the loss of technical support from the WRS, as well as financial constraints and socio-political difficulties in Haiti. The data time series and a description of the methods used to collect them were published annually in the "Hydrographic Bulletin", summarizing 70 daily streamflow time series over the 1920-1940 period. After these two decades of streamflow observations, very few hydrological data were produced in Haiti (Pouyaud and Hoepffner, 1987). In addition to hydrometric observations, rainfall measurements started in Haiti around 1905 using 15 raingauges. Over time, the raingauge network became denser, with 25 stations operated by the "Petit-Séminaire Collège St Martial" (a school run by the Congrégation du Saint-Esprit), 38 by the "Direction Générale des Travaux Publics", and nearly 30 by other institutions, such as the" Frères de l'Instruction Chrétienne" (Pouyaud and Hoepffner, 1987). Rainfall measurements are currently managed by the CNIGS and the UHM. Since 2014, this observation network has had approximately twenty automatic raingauges. However, due to a significant amount of missing data, the network remains highly fragmentary and unexploited.*

*In 1977, the Haitian government initiated a project to make an inventory and digitize some available hydroclimatic time series. As a result, the 70 daily streamflow series for the period 1920-1940 and almost a hundred monthly rainfall series from the start of observations (~1905) until 1975 were digitized. In 2012, the Haitian government launched a second project named BVH (Bassins Versants Haïtien in French, i.e., Haitian catchments; Gaucherel et al. 2018) for compiling available hydroclimatic data, better understanding hydrology in Haiti and improving the management of water resources. Within this project, Haitian catchments were characterized using monthly streamflow data (Gaucherel et al., 2016) and rainfall data (Moron et al., 2015) and the relationships between their shape, relief, and river sinuosity were investigated (Gaucherel et al., 2017, Bonhomme et al., 2013).Unfortunately, the two databases produced within the BVH project (monthly rainfall time series and monthly streamflow time series) have never been analyzed jointly, are not available online and remain limited for several hydrological analysis due to their monthly time step (monthly). Thus, these databases are underused to date.*

b.  *Is there a schedule for digitizing the more recent observed (runoff) data? Are all (or at least the most) gauges still in operation? If so, it would be very nice to do the work in the context of this paper and update your results. I think that more recent results are more relevant than those from the period 1920 to 1940.*

After 1940, there is a scarcity of streamflow observation data available in Haiti. Good quality streamflow data can only be found for the period between 1920-1940, which has been presented in Simbi.

See also responses to comment a.

*c. The uncertainty of the results is in some parts high (KGE values < 0,5 are often referred to as poor results). I would add a chapter "Uncertainties" in or after the results to address and describe all sources of uncertainties (e.g. long time ago, measurement method, data aggregation, model). I am aware that uncertainties are omnipresent with such a data basis, but from my point of view it would be helpful to list them clearly and centrally at one place in your manuscript.*

Thanks for this suggestion, an uncertainties section has been added in section 5 to the revised manuscript.

*5 Uncertainties*

*This section discusses the main sources of uncertainty associated with the Simbi database. These uncertainties can be classified into four main types:*

*1. The Simbi database contains historical data, which may be prone to errors due to factors such as the used equipment, methods employed to measure flows, and the establishment of rating curves. For most streams, water levels were measured manually by reading a vertical scale placed on one of the banks of the stream two or three times a day. Over time, 12 automatic recorders have been installed on 11 rivers, providing automatic and continuous readings of water levels on these streams. The metadata indicates the quality of the rating curves and the stations where the automatic recorders have been installed.*

*2. The historical data was originally in paper format and has been digitized. Despite quality control tests, uncertainties remain regarding the digitization of paper archives.*

*3. The raingauges identified as relevant for hydrological modelling in this article depend on the use of a rainfall-runoff model. Different methodologies or rainfall-runoff models may produce different results and thus different catchment-scale precipitation forcing.*

*4. The simulated streamflows are dependent on the rainfall-runoff models that are used and may differ if other models are used. It is important to note that, especially at the daily time step, the KGE obtained for some catchments are poor (KGE<0.5).*

*d. In addition, I would make a very clear statement at the end and also in your abstract about the high uncertainties. Otherwise, there is the risk that further working groups do not sufficiently consider the high uncertainties.*

The uncertainties have been clearly indicated in the abstract and the revised manuscript.

*Haiti, a Caribbean country, is highly vulnerable to hydroclimatic hazards due to heavy rainfall, which is partly linked to tropical cyclones. Additionally, its steep slopes generate flash floods, particularly in small catchments. Moreover, the hydrology of this region remains poorly understood and understudied. Unfortunately, there is no accessible database for the scientific community to use in this country. To fill this gap, hydroclimatic data were collected to create the first historical database in Haiti. This database, called "Simbi" (guardian of rivers, freshwater, and rain in Haitian mythology), includes 156 monthly rainfall series over the period 1905–2005, 59 daily rainfall series over the period 1920–1940, 70 daily streamflow series, and 23 monthly temperature series, not necessarily continuous, over the period 1920–1940. It also provides simulated streamflow series over the period 1920–1940 using the GR2M and GR4J rainfall–runoff models for 24 catchments and 49 attributes covering a wide range*

*of topographic, climatic, geological, land use, hydrogeological, and hydrological signature indices. Simbi is the first open-access hydro-meteorological dataset for Haiti and will contribute to a better knowledge of hydrological risk in Haiti. Several sources of uncertainty associated with Simbi are acknowledged, including data quality (historical data), digitisation of paper archives, identification of relevant raingauges, and rainfall-runoff models. It is important to consider these uncertainties when using Simbi.*

e. *The used data sources (e.g. land cover) and method for calculation the catchment attributes are not always clearly specified and listed. This is a lack of transparency.*

Section 3.5 has been revised to include the data sources used and the method for calculating catchment attributes.

[revised manuscript text omitted]

> f. *Try to specify the data products used for calculating your catchment attributes (e.g. year of publication, domain, spatial resolution, update frequency, reference) at least in a list in the appendix.*

Information on the data used is provided in Table 2 (See response to comment e).

> g. *The structure of your dataset isn´t friendly for machine-reading. Details are listed below.*

The structure of the dataset has been modified for machine-reading (see to the dataset section below).

> h. *Sharing the codes is helpful for reproducing your results and would make it easier for other groups to build upon your work.*

The file SIMBI_code_R.r file contains an R script for importing data, calculating catchment scale data and calibrating the GR2M model, has been added to the database.

> i. *Links should be listed in the references and not in the text, in order not to disrupt the reading flow.*

The links have been removed from the text, except for the link in the abstract because the dataset link must be included in the abstract according to the ESSD manuscript preparation guidelines.

3. **Minor comments**

> *They are just a first glimpse for the start, see clarification.*

> a. *L45: The CAMELS datasets also provide indices and signatures.*

The sentence has been replaced by

*While the CAMELS databases provide time series, indices and hydroclimatic signatures of catchments, other databases provide only indices and hydroclimatic signatures of catchments, such as the African Database of Hydrometric Indices (ADHI; Tramblay et al., 2021).*

> b. *L111: Can the digitization process be described (e.g. with an indication of accuracy)?*

The digitization process has been described in section 2.2.2

[revised manuscript text omitted]

*Hereafter, we will only use areas calculated with TauDEM algorithm and not areas noted in the paper archives. The geographic locations of the 24 selected hydrometric stations are shown as red dots in Figure 2.*

*d. L181: What Is indicated by the white, orange and blue dots? A legend would make it much easier to understand the figure.*

The white, orange and blue dots indicated all raingauges with monthly data for the period 1920–1940. Raingauge stations with air temperature data are shown in orange. Raingauges considered relevant for hydrological modeling are shown in blue (see section ***Erreur ! Source du renvoi introuvable.***). Other raingauges are shown in white. A legend has been added in the revised manuscript.

[Figure]

*Figure 2 – Location of the 24 hydrometric stations used (red dots), the associated catchment contours (black solid lines), and the location of all raingauges with monthly data for the period 1920–1940 (white, orange, and blue dots). Raingauge stations with air temperature data are shown in orange. Raingauges considered relevant for hydrological modeling are shown in blue. NOAA 20CR pixels are shown in purple, the border between Haiti and the Dominican Republic is shown as a dashed black line, and the background topography is from the SRTM database.*

*e. L238: Why were the two individual datasets NOAA 20CR and BEST area-weighted averaged? Can a dataset be used without the other one?*

Both air temperature databases were used independently. The wording of the previous sentence was unclear, so it has been revised:

*Catchment air temperature series were computed at a daily time step for two temperature databases (NOAA 20CR and BEST) by taking the weighted average of pixels in the respective database (NOAA 20CR or BEST).*

*f. L273: How was the aggregation done (e.g. area-weighting of the intersecting pixel or weighted all pixels equal not matter of the intersected individual area) and what was the output (e.g. mean, median). This information would be important.*

This section (section 3.5) has been revised. See response to "main comments (e)"

*g. L276: What is relevant to Haitian catchments and what not? I think the most readers do not have this information.*

Thanks for this suggestion. This sentence will be reworded:

*Similar to the CAMELS databases (Addor et al., 2017, Alvarez-Garreton et al., 2018, Chagas et al., 2020, Coxon et al., 2020, Fowler et al., 2021, Klingler et al., 2021), a set of attributes that describes a broad range of low, moderate and high precipitation and streamflow characteristics were chosen to characterize the hydrological regime of each catchment.*

h. *L279: Did you use the R-codes of Nans Addor (CAMELS-US) to calculate the hydrological signatures? If yes, it would be good to mention that.*

We did not use the R codes of Nans Addor to calculate hydrological signatures.

i. *L287: It seems that the two datasets for land cover are different data sources (other attributes). Can you give details about the differences (e.g. data genesis).*

This section (section 3.5) has been revised. See response to "main comments (e)"

j. *L288: There are global land use datasets (mostly derived by earth observation methods) for the most recent period.*

This section (section 3.5) has been revised. See response to "main comments (e)"

k. *L293: What is meant with "cut out"?*

The expression "cut out" will replaced by "cropped".

l. *L367: What is the red line and the red point indicating in the figure?*

The red line represents the optimal ratio (r=1), while the red dot represents the mean value of the distribution. The figure caption has been modified:

*Figure 8 – Ratio of the GR2M calibrated parameters X1 (a) and X2 (b) over the two subperiods for the reference and relevant raingauge combinations. The red line represents the optimal ratio (r=1), while the red dot represents the mean value of the distribution.*

m. *L376: Try to be more precise, what is a critical percentage of missing data?*

If a station has less than 10 years of data, it will be deemed to have a critical percentage of missing data (The 10-year threshold was set arbitrarily). This paragraph has been revised

*Figure 9 shows that the raingauges used for the relevant raingauge combinations are those located at low elevations and with the longest data series. The relatively low percentage of missing data from the relevant raingauges ensured better model stability (see section 4.2.2) and contributed to the improvement in the model performance, especially by reducing the biases between simulated and observed streamflow (improvement in α and β parameters; see section 4.2.1). Raingauges at higher elevations are more difficult to access and are the least maintained, and therefore have very high percentages of missing data (raingauges with less than 10 years of data). However, the model tends to discard raingauges with high percentages of missing data, which is why the retained/selected raingauges are generally located at lower elevations. There is no clear trend of monthly rainfall in the selection of relevant raingauges. However, some very wet raingauges (rainfall totals over 180 mm/month) were selected as relevant raingauges.*

n. *L392: If the relevant raingauge combination can cause such an error, have we to conclude that this amount of deviation is also possible at the other catchments? By the way: Can you declare the weights of the applied raingauge combination in a text file?*

If the relevant rain gauges are wet or dry, a greater or lesser deviation may be observed. The weights of the raingauge combination has been added (05_SIMBI_WEIGHTS_RAINGAUGE folder) in the revised database.

o. *L423: There is partly a big difference of mean annual discharge between neighbouring gauges (> 600 mm/year). Can you explain that?*

The Q-008 watershed has a significantly higher mean annual streamflow than the three adjacent watersheds (Q-010, Q-068 and Q-029). As indicated in line 387, over 90% of the Q-008 watershed is located on a limestone geological formation, and there is also an impact from karst aquifers within the watershed. On this basis, it is likely that an influx of water from neighboring catchments is responsible for such a high mean annual streamflow. However, no studies have been carried out to verify or challenge this hypothesis. This point has been highlighted in the revised manuscript.

*The observed and simulated mean annual streamflows from GR2M are illustrated in Figure 12. The results show that streamflow is higher in the southwest and north of Haiti and lower in the central part. However, the Q-008 catchment has a significantly higher mean annual streamflow than its three neighboring catchments (Q-010, Q-068 and Q-029). As shown in section 4.3, over 90% of the Q-008 catchment is situated on a calcareous geological formation, 40% of which is on karstic aquifers. Therefore, it is probable that an influx of water from neighboring catchments is accountable for such a high mean annual streamflow. Nevertheless, no study has been conducted to confirm or dispute this hypothesis. The simulated streamflow represents well the spatial pattern of the observed streamflow and gives good estimates of the observed mean annual streamflow.*

p. *L470: This sentence fits better in the conclusion.*

Thanks for this suggestion. This sentence has been moved in the conclusion

q. *L484: How where the return periods calculated? Which statistical distribution?*

The Generalized Extreme Value distribution and the distribution of annual values (precipitation, PET, air temperature or streamflow) were used to estimate values for multiple return periods. This sentence has been clarified in the revised manuscript:

*Simulated streamflows underestimate maximum annual flows with a return period of less than 10 years and overestimate flows beyond 10 years. The generalized extreme value (Beirlant et al., 2004; Coles, 2001; Jenkinson, 1955) and the distribution of annual values (precipitation, PET, air temperature or streamflow) were used to estimate values for multiple return periods.*

r. *L492: I would set the chapter Data availability after Conclusion.*

The data availability section must be placed before the conclusion section according to the ESSD manuscript preparation guidelines.

s. *L518: There are also outliers and not really a clear trend, see Figure 9.*

Thanks for this comment. Indeed, the central part of Haiti is associated with low streamflow values and the southwest with high streamflow valuers. However, no clear trend was observed in the north. These sentences have been revised:

*The central part of Haiti is associated with relatively low streamflow and high drought coefficients. The southwest is associated with relatively high streamflow. In fact, large floods are more frequent in these areas (Terrier et al., 2017). No clear trend was observed in the north.*

t. *L600: An additional column with the source dataset would enhance transparency in Table C1, C2 and C3.*

An additional column with the source dataset has been added in the table in the appendix in the revised manuscript (see also table 2).

4. **Dataset**

*a.* *It is not user-friendly, that your dataset is compressed 2-fold. 1-fold (all files together) is enough.*

Thank you for your comment. Unfortunately, this two-time compression is due to the data warehouse that adds extra folders, and we are not able to change that.

*b.* *I would name the compressed file Simbi and not "dataverse_files".*

The folder "dataverse_files" is added by the data warehouse and is not reliant on us.

*c.* *The dataset is unnecessarily nested after extracting (e.g. dataverse_files\SIMBI\SIMBI_00_OBSERVED_DATA\00_OBSERVED_DATA\). Folder levels could be removed.*

The first three levels (dataverse_files\SIMBI\SIMBI_00_OBSERVED_DATA) are automatically included by the IRD data warehouse and therefore cannot be removed. These directories are intended for use by other databases also hosted within this warehouse, such as the ADHI database (https://doi.org/10.23708/LXGXQ9).

*d.* *Storing the files as .csv instead .txt would enable an easy and fast opening with the software Excel. This is nice for a quick view.*

The files have been saved in .csv format.

*e.* *The files are not friendly for machine-reading. I would remove the description in the header of your data files and create a new file with the belonging metadata. In this metadata file it is recommended to list the important information (e.g. Station-code, unit) directly after the name of the variable without blanks or special characters, e.g.: Station_code;P-001 Unit;mm/d → Hint: Get rid of the blanks in the metadata, they are good for human reading, but not for machine reading.*

The headers of the data files have been deleted and new files have been created to store the metadata. The metadata files include:

   i.   a file containing information on rain gauge stations,
   ii.  a file containing information on streamflow stations,
   iii. a file containing monthly time-step rainfall-runoff modeling information, and
   iv.  a file containing daily time-step rainfall-runoff modeling information.

*f.* *It is tedious to read and separate lines like line 10 in Q_001___AMONT_DU_BASSIN___RIVIERE_GRISE (## Station code : 40501 ; Q-001 ; (DHSI ; URGEO). Try to get the DHSI station code (40501) with R, then you see why. In general, try to make it as easy as possible to import data and the belonging metadata e.g. with the software R.*

Metadata has been stored in separate files in a tabular format, allowing for easy importation.

*g.* *I would rename the files with the timeseries (e.g. instead Q_001___AMONT_DU_BASSIN___RIVIERE_GRISE → Q_1)*

Files have been renamed using station codes to facilitate identification (e.g. instead of Q_001___AMONT_DU_BASSIN___RIVIERE_GRISE → Q_001).

*h.* *I would indicate the gaps in the time series not with a character (NA), simply do not enter anything. This ensures the data format numeric.*

Blank spaces are handled differently depending on the programming language and version used. To ensure ease of use for all users, including those working with R, Python, Excel, and other software, missing values are represented by -9999, as is common practice in other databases (mainly climate databases). This has been explained clearly in the manuscript and the dataset.

*i.    The ID of the gauge (code) should always be in the first column (this is not the case e.g. in the file location_and_topography.txt).*

Thanks for this comment. Station IDs are now in the first column.

*j.    The number of decimal places is sometimes really too high (e.g. Percent_geologic_class.txt). This implies an accuracy which is not given.*

Thanks for this comment. The decimal places are now limited to 2.

*k.    The stream density includes a lot of information of a catchment (e.g. hint for carstic region). I would calculate and add this important catchment attribute. If there is no local dataset with the Haitian streams, take a global dataset like HydroATLAS (Linke et al., 2019, https://doi.org/10.1038/s41597-019-0300-6)*

Thanks for this suggestion. Stream density has been included as an attribute in the revised manuscript by utilizing dataset from CNIGS (National Center for Geospatial Information in Haiti).

*l.    Have a look at Gudmundsson et al. (2018, https://doi.org/10.5194/essd-10-787-2018) regarding quality control flags (e.g. value does not change over more than x timesteps, outlier) for your timeseries.*

Thanks for this suggestion. In addition to the criteria outlined in section 3.1.1 for selecting streamflow series, the three indicators proposed by Gudmundsson et al. (2018) for quality control of time series have been added in the revised manuscript:

      i.   days for which Q<0, where Q denotes a daily streamflow value,

     ii.   daily values with more than 10 consecutive equal values larger than zero, and

    iii.   outlier detection.

The section 3.1.1 has been revised.

*3.1.1 Selection of streamflow series*

[revised manuscript text omitted]

*m. Adding metadata to your shapefiles would be helpful, where the header and the unit of your attributes is declared.*

metadata (name, code, altitude and area) and their units will be added to the shapefiles.

**5. Plots**

   *a. The colour scales of your plots are not always intuitive (e.g. Figure 4 or 9). There is a nice article from Stoelzle and Stein (2021, https://doi.org/10.5194/hess-25-4549-2021) about visualization in hydrology. I recommend considering this guidance for your figures.*

   Thank you for this comment. The revised manuscript includes modified color scales for the figures.

   *b. The font sizes in your plots vary sometimes greatly (e.g. Figure 7). It would be good to homogenize the sizes.*

   The font sizes of the plots have been homogenized in the revised manuscript.

   *c. It is often more intuitive for the reader to extend the legend of a plot, than describing the attributes in the label of the figure (e.g. Figure 1).*

   Figure attributes has been described in the figure legend in the revised manuscript.

**B – ANSWER TO THE REVIEWER #2**

1. **ABSTRACT**

   a. *Besides being "exposed to hydroclimatic hazards" is there anything else that would make the hydrology of the area interesting? The paper could clarify this point starting from the abstract.*

   It is a region with rugged slopes that favor flash floods, especially in small catchments. Additionally, the hydrology of this region is poorly understood and understudied.

   The changes in the manuscript are:

   *Haiti, a Caribbean country, is highly vulnerable to hydroclimatic hazards due to heavy rainfall, which is partly linked to tropical cyclones. Additionally, its steep slopes generate flash floods, particularly in small catchments. Moreover, the hydrology of this region remains poorly understood and understudied. Unfortunately, there is no accessible database for the scientific community to use in this country.*

   b. *What are the limitations of the dataset? Why is it that mainly the 1920-1940 period is analyzed?*

   Reviewer 1 recommends dedicating a section to database uncertainties, which has been summarized in the abstract and detailed in section 5.

   The changes in the abstract are:

   *Simbi is the first open-access hydro-meteorological dataset for Haiti and will contribute to a better knowledge of hydrological risk in Haiti. Several sources of uncertainty associated with Simbi are acknowledged, including data quality (historical data), digitisation of paper archives, identification of relevant raingauges, and rainfall-runoff models. It is important to consider these uncertainties when using Simbi.*

2. **INTRODUCTION**

   a. *P1L45 – CAMELS also provides hydroclimatic signatures?*

   The sentence has been replaced by

   *While the CAMELS databases provide time series, indices and hydroclimatic signatures of catchments, other databases provide only indices and hydroclimatic signatures of catchments, such as the African Database of Hydrometric Indices (ADHI; Tramblay et al., 2021).*

   b. *P2L47 – Define "these works".*

   "These works" is replaced by "this study"

   The changes in the manuscript are:

   *The main objectives of this study are to make Haitian hydroclimatic data available to the scientific community and to merge these different datasets in order to propose the first hydroclimatic database for several Haitian catchments at both monthly and daily timesteps.*

   c. *P2L50 – It reads like there is nothing for Haiti, but it seems that the dataset was built on previous data. Shouldn't the background be provided here?*

   Thanks for the comment. This article relies on data from the BVH project conducted in Haiti in 2012.

   The changes in the introduction to the revised manuscript are:

   *Unfortunately, there are significant differences between countries in terms of the quality and quantity of hydroclimatic reference databases, as well as regarding access to these data. Some countries do not have such reference databases. This is the case of Haiti, whose territory is, moreover, highly exposed to natural disasters (Khouakhi et*

*al., 2017; Burgess et al., 2018), and climate change (Peterson et al., 2002). At the same time, Haiti is facing the consequences of massive deforestation and anarchic urbanization (urban development that does not comply with planning regulations) in recent decades (Hedges et al., 2018; Tarter et al., 2018; Mompremier et al., 2022), resulting in increased vulnerability to hydroclimatic hazards. Currently, Haiti lacks a freely and easily accessible hydroclimatic database due to the absence of in situ hydroclimatic observations. The first hydrometric observations were conducted during the American occupation of Haiti, and began in 1919. American engineers from the Water Resources Service (WRS) of the United States Geological Survey (USGS) supervised these hydrological observations, that continued into the 1940s and exceptionally later. The end of the American occupation is the main reason for the cessation of hydrometric observations. This is due to the loss of technical support from the WRS, as well as financial constraints and socio-political difficulties in Haiti. The data time series and a description of the methods used to collect them were published annually in the "Hydrographic Bulletin", summarizing 70 daily streamflow time series over the 1920-1940 period. After these two decades of streamflow observations, very few hydrological data were produced in Haiti (Pouyaud and Hoepffner, 1987). In addition to hydrometric observations, rainfall measurements started in Haiti around 1905 using 15 raingauges. Over time, the raingauge network became denser, with 25 stations operated by the "Petit-Séminaire Collège St Martial" (a school run by the Congrégation du Saint-Esprit), 38 by the "Direction Générale des Travaux Publics", and nearly 30 by other institutions, such as the" Frères de l'Instruction Chrétienne" (Pouyaud and Hoepffner, 1987). Rainfall measurements are currently managed by the CNIGS and the UHM. Since 2014, this observation network has had approximately twenty automatic raingauges. However, due to a significant amount of missing data, the network remains highly fragmentary and unexploited.*

*In 1977, the Haitian government initiated a project to make an inventory and digitize some available hydroclimatic time series. As a result, the 70 daily streamflow series for the period 1920-1940 and almost a hundred monthly rainfall series from the start of observations (~1905) until 1975 were digitized. In 2012, the Haitian government launched a second project named BVH (Bassins Versants Haïtien in French, i.e., Haitian catchments; Gaucherel et al. 2018) for compiling available hydroclimatic data, better understanding hydrology in Haiti and improving the management of water resources. Within this project, Haitian catchments were characterized using monthly streamflow data (Gaucherel et al., 2016) and rainfall data (Moron et al., 2015) and the relationships between their shape, relief, and river sinuosity were investigated (Gaucherel et al., 2017, Bonhomme et al., 2013).Unfortunately, the two databases produced within the BVH project (monthly rainfall time series and monthly streamflow time series) have never been analyzed jointly, are not available online and remain limited for several hydrological analysis due to their monthly time step (monthly). Thus, these databases are underused to date.*

    d.   *P2L53 – What is the definition of "anarchic urbanization"?*

Anarchic urbanization is the practice of urbanizing in a way that disregards town planning regulations.

The changes in the manuscript are:

*At the same time, Haiti is facing the consequences of massive deforestation and anarchic urbanization (urban development that does not comply with planning regulations) in recent decades (Hedges et al., 2018; Tarter et al., 2018; Mompremier et al., 2022), resulting in increased vulnerability to hydroclimatic hazards.*

    e.   *P2L63 – "These two databases…" See previous comment on line 50.*

The two databases are the monthly rainfall database (1905-2005) building by Moron et al. (2015) and the 70 daily streamflow series (1920-1940) from the BVH project.

See responses to comment d.

*f. P2L75 – Is "producing" the appropriate verb here?*

"producing" is replaced by "building".

The changes in the manuscript are:

*ii) building climatic (air temperature and rainfall) time series at the catchment scale by spatially and temporally aggregating available series,*

*g. P2L77 – Why for the 1920-1940 period?*

The of 1920-1940 period is chosen as it is the only time period with high-quality streamflow measurements available.

See the changes in the comment d.

3. **DATA USED**

*a. L90 – What is the meaning of 70 daily series, but 24 are used? Why 1920-1940? Provide a better description of missing data etc.*

70 streamflow series are available, with 24 selected for rainfall-runoff modeling based on their quality. The process for selecting streamflow is explained in section 3.1.1.

The section 3.1.1 has been revised

*3.1.1 Selection of streamflow series*

*An analysis of the 70 available streamflow series was performed to select the "hydrologically relevant" streamflow series. Four criteria were initially used to make this selection:*

*1. The annual hydrographic bulletins reported the accuracy with which rating curves were established through three ratings: "well established," "fairly well established," and "poorly established." Most of the streamflow series with "poorly established" rating curves were found to have significant measurement differences between periods. These streamflow series were not used in the remainder of this study.*

*2. Some hydrometric stations were located downstream of diversion channels or small dams used for irrigation. These streamflow series poorly represent the seasonality of streamflow, and are therefore considered to be influenced by human activities. These streamflow series were not used in the remainder of this study.*

*3. Some hydrometric stations were located downstream of resurgences or springs. These groundwater resurgences are beyond the scope of this study. Therefore, these streamflow series were not used in the remainder of this study.*

*4. The streamflow series that had less than 5 years of data were not used in the remainder of this study.*

*In addition to these four criteria, three other indices inspired by the paper of Gudmundsson et al. (2018) were used to assess the quality of the streamflow data. These three criteria were calculated as follow:*

*1. Number of days for which Q<0, where Q denotes a daily streamflow value. The rationale underlying this rule is that streamflow values smaller than zero are non-physical (Gudmundsson and Seneviratne, 2016).*

*2. Sequence of more than 10 equal consecutive streamflow values larger than zero. This index was selected because equal consecutive streamflow values often occur due to instrument failure or flow regulation (Gudmundsson et al., 2018).*

*3. Detection of outliers, i.e. unusually large or small streamflow values that could come from instrument malfunction. The calculation of these outliers is inspired by the papers of Gudmundsson et al. (2018): daily streamflow values are flagged as outliers if values of*

*log (Q+0.01) are larger or smaller than the mean value of log (Q+0.01) plus or minus 6 times the standard deviation of log (Q+0.01) computed for that calendar day over the entire series. The mean and standard deviation are computed for a 5-day window centered on the calendar day to ensure that a sufficient amount of data is considered. The log-transformation is used to account for the skewness of the distribution of daily streamflow values and 0.01 was added because the logarithm of zero is undefined.*

*To summarize, the quality of the 70 streamflow daily series is described using 12 flags (1, 2, 3, 4, A, B, C, D, E, F, H and I), as detailed in the Table 1. Using these criteria, along with visual analysis to identify anomalies (i.e. non-natural records that may be erroneous streamflow values or anthropogenic influences that can lead to misinterpretation of actual hydrological processes (Strohmenger et al., 2023)), 24 hydrometric stations were identified as "hydrologically relevant" from the 70 available.*

b.  *L106 – Can you also provide a graph showing missing data periods and percentage?*

The percentages of missing data have been quantified and presented graphically in revised manuscript.

[Figure]

*Figure 1 – Period of availability and percentage of stations with data available for digitized daily rainfall datasets, daily streamflow datasets, monthly air temperature datasets, and monthly rainfall datasets produced by Moron et al., (2015).*

c.  *L122 – Exactly what significant gaps?*

"Significant gaps" is replaced by "missing data"

4.  **METHODOLOGY**

a.  *L138 – Why for 21 of 24?*

24 streamflow series were selected from a total of 70. Monthly rainfall-runoff modeling was performed on those 24 associated catchments. However, only 21 of the 24 catchments had available daily rainfall series. Consequently, daily rainfall-runoff modeling was performed for these 21 catchments.

this sentence is better explained and moved to section 3.4:

*The GR2M model was used to simulate the monthly streamflow series for the 24 catchments studied, and the GR4J model was used to simulate the daily streamflow series for 21 of the 24 catchments where daily rainfall data are available.*

*b. L159 – Why were the remainder not provided?*

These streamflow series were not selected because they are located downstream of diversion canals or small dams used for irrigation. They poorly represent streamflow seasonality and are therefore considered to be influenced by human activities. This paragraph has been reworded in revised manuscript (see also response to the comment "data used a").

*2.        Some hydrometric stations were located downstream of diversion channels or small dams used for irrigation. These streamflow series poorly represent the seasonality of streamflow, and are therefore considered to be influenced by human activities. These streamflow series were not used in the remainder of this study.*

*c. L169 – What is the meaning of "numerically calculated"?*

Catchment contours were delineated using algorithms (digitally) as opposed to contours estimated using topographic maps in historical times. The section 3.1.2 has been revised.

*3.1.2 Catchment boundaries and areas*

*The contours of the 24 catchments corresponding to the 24 selected hydrometric stations were delineated using the SRTM digital terrain model (Reuter et al., 2007) and the TauDEM algorithm (Tarboton et al., 2005). The catchment areas calculated with TauDEM algorithm were compared with those reported in the "Hydrographic Bulletin" (areas estimated from U.S. Army maps). Table 1 in Appendix C presents the ratios and errors between the areas calculated with TauDEM and those in the hydrographic bulletins. The errors between the two areas are less than 10% for 18 of the 24 catchment areas (blue dots). However, significant errors were observed for 6 catchments (Q-045, Q-051, Q-056, Q-060, Q-061 and Q-065). Three factors account for significant differences between the two areas:*

*1.        The positions of some hydrometric stations were wrong in the archives. Their locations were corrected using additional information in the hydrographic bulletins (name of a bridge, main road, monuments, etc.). For example, the name of a bridge for station Q-056 (Pont Parois) and the name of the river for station Q-060 (Massacre river) were used to correct the station position*

*2.        Due to the low resolution of the DEM, the river network generated with TauDEM algorithm may differ from the real river network, especially in plain areas near the estuaries. Hydrometric stations were therefore relocated to match the stream generated by the TauDEM algorithm (stations Q-045, Q-065 and Q-051).*

*3.        Three different stations (Q-053, Q-061 and Q-056) were associated to an upstream catchment area equal to 252 km².We supposed that this is an error in the areas of the hydrographic bulletins.*

*Hereafter, we will only use areas calculated with TauDEM algorithm and not areas noted in the paper archives. The geographic locations of the 24 selected hydrometric stations are shown as red dots in Figure 2.*

*d. L171-172 – What does it mean to "relocate manually"?*

The provided geographical locations for certain stations did not correspond to the information in the hydrographic bulletins. The locations of these stations were rectified utilizing the information obtainable in the hydrographic bulletins. This paragraph has been revised (see previous response (c)).

*e. L174 – What "numerical model"?*

The "numerical model" is the DEM (digital elevation model).

See the changes in the comment c.

*f.* *L177 – This is not clear.*

Rivers generated with SRTM differ from real rivers. Station positions must be relocated to match the rivers generated by SRTM. This paragraph describes the station relocation process.

See the changes in the comment c.

*g.* *FIG.1 – Provide Haiti country boundary. Are the gray lines the river network? How to consider if a raingauge is relevant? What are the white dots?*

Gray lines indicate the river network; white dots represent non relevant raingauges. Relevant rain gauges are defined in section 3.2.1.

the figure is moved to the end of section 3.2.1. The changes in the manuscript are:

[Figure]

*Figure 2 – Location of the 24 hydrometric stations used (red dots), the associated catchment contours (black solid lines), and the location of all raingauges with monthly data for the period 1920–1940 (white, orange, and blue dots). Raingauge stations with air temperature data are shown in orange. Raingauges considered relevant for hydrological modeling are shown in blue. NOAA 20CR pixels are shown in purple, the border between Haiti and the Dominican Republic is shown as a dashed black line, and the background topography is from the SRTM database.*

*h.* *L187 – Is this the best method to identify appropriate raingauge?*

This method seems to be the most appropriate for identifying the relevant raingauges for rainfall-runoff modelling. Indeed, the performance of a rainfall-runoff model improves with a better description of the rainfall input (Andréassian et al., 2001). The GR2M model was therefore used to determine the relevant raingauges in this study. Relevant raingauges are defined as those with the best performance of the model.

this paragraph has been better explained in the revised manuscript. The changes in the manuscript are:

*The performance of a rainfall-runoff model improves with a better description of the rainfall input (Andréassian et al., 2001). The GR2M monthly rainfall-runoff model was*

*therefore used to determine, for each catchment and at the monthly timestep, the "relevant" raingauges in this study. NOAA 20CR rainfall series, reference rainfall series and multiple raingauge combinations are used as inputs to the GR2M model and relevant raingauges are defined as those providing the best model performance.*

*The first 3 years of data (early 1920 to late 1922) were used to initialize the model, and a split-sample test (Klemeš, 1986), commonly used in hydrology, was implemented. This practice consists in splitting a streamflow time series into two distinct subperiods P1 and P2, the first for calibration and the second for evaluation, and then exchanging these two subperiods. The two subperiods P1 and P2 are chosen so that they have the same available streamflow lengths. The combination of raingauges with the best KGE score in evaluation (average of the KGE in evaluation over the two subperiods) was considered as the most relevant for rainfall–runoff modeling.*

i.  *L213-214 – Can this be shown in a figure? What is a continuous catchment rainfall?*

Continuous catchment rainfall is a catchment-scale rainfall series with no missing data.

The changes in the manuscript are:

*All possible raingauge combinations are calculated for each catchment (combination of 1, 2, 3,…, n raingauges, where n is the number of available raingauges). If a single raingauge is available, its data is used as the catchment scale rainfall series (weighting coefficient = 1). If there are multiple raingauges available, their weighting coefficients are calculated from the Thiessen polygons. Catchment scale rainfall series with no missing data were used for rainfall–runoff modeling.*

j.  *L216-221 – This is not clear.*

This paragraph describes the procedure for calibrating and evaluating the performance of the GR2M rainfall-runoff model over two distinct sub-periods, P1 and P2, for each catchment.

section 3.2.1 has been reworded. See the changes in the comment h.

k.  *L222-224 – Why was it used then? This is contradictory to the previous explanation.*

This is not in contradiction with the previous explanation. This paragraph presents the limitations of the method used and explains the dependence of the results on the rain-streamflow model used.

this paragraph has been moved to section 5 dedicated to uncertainties.

*3.      The raingauges identified as relevant for hydrological modelling in this article depend on the use of a rainfall-runoff model. Different methodologies or rainfall-runoff models may produce different results and thus different catchment-scale precipitation forcing.*

l.  *L253 – What is the meaning of "most relevant method"?*

This sentence has been reworded as follows:

*Moreover, it is one of the most relevant approach for rainfall–runoff modeling compared to 27 models for calculating PET and has been tested on more than 300 catchments covering several climatic zones, including tropical zones (Oudin et al., 2005).*

m.  *L263 – Parameters are called "Period 1"… this is not clear and not the best choice of terms.*

The three periods used for calibration will be referred to as P1, P2 and P3 in the next version of the manuscript.

n.  *L277 – SRTM and catchment attributes should be all explained in the same section.*

Section 3.5 has been revised. To improve clarity, the 'location and topography' attributes are explained in the same section as the other attributes.

The changes in the manuscript are:

*3.5.1    Location and topography attributes*

*Table C3 presents the six location indices that were calculated. Catchments are identified by the same codes as the hydrographic stations, in the format Q-XXX, where XXX ranges from 001 to 070 to identify the 70 hydrographic stations. The catchments have the same names as the hydrographic stations and are taken from the hydrographic bulletins. The longitudes and latitudes of the outlets correspond to those of the hydrometric stations presented in section 3.1.2 (and includes coordinates modification). The longitudes and latitudes of the catchment centroids were calculated based on the catchment contours delineated in section 3.1.2.*

*The topographic attributes include area, elevation, slope, catchment elongation, and drainage density. Catchment areas were calculated using the SRTM digital terrain model and the TauDEM algorithm (see section 3.1.2). Elevation is a key factor in hydrological processes as it influences many other catchment characteristics (Addor et al., 2017). Therefore, minimum and maximum elevations, standard deviations, hypsometric curves (empirical elevation distribution function) and average catchment slopes were calculated using the SRTM digital terrain model. The average slopes of the catchments were calculated using the SRTM digital terrain model and the algorithm of Horn (1981). The Gravelius index, which provides information on the elongation of the catchment and therefore influences the hydrograph, was calculated. The Gavelius index is defined as the ratio of the perimeter of the catchment to the circumference of a circle with the same area (Bendjoudi and Hubert, 2002). Finally, stream density, the ratio of the total of all stream segments to the area of the catchment, was calculated using the CNIGS (Centre National de l'Information Géospatiale in French i.e national center for geospatial information) river network shapefile. The stream density is influenced by the density of the hydrographic network and therefore by the permeability of the catchment.*

o. *L283 – to produce… are produced.*

Section 3.5 has been revised and this sentence has been deleted in the revised manuscript.

p. *L288-292 – Should be clearly explained/justified in the beginning. I figure showing the changes would be interesting.*

This paragraph has been better presented in the revised manuscript, taking into account your comments and those of reviewer 1.

The changes in the manuscript are:

*3.5.4    Land cover*

*Land cover data for Haiti is provided by the CNIGS and is only available for two periods: 1995 and 1998.*

*Although the land cover classifications used in 1998 differ from those used in 1995, Figure 3 illustrates that most of the woodland areas in 1995 were converted to cropland, grassland, or savannah in 1998. According to the 1998 classification, medium-density cropland is the most dominant land use, accounting for a quarter of the total the territory. High-density agro-forestry systems occupy 18%, high-density agricultural crops 17%, savannah 7.3%, pasture with other uses 4.7%, wetlands 4.4%, rock outcrops and bare ground 1.8% and forest 1.25%. The area of other types of use is generally less than 1% of the territory.*

*Shapefile of land cover data (1995 and 1998) were cropped for each of the catchments studied. The proportion of each land cover class occupied in the catchment was then calculated, corresponding to the two land cover indices calculated in Simbi: "cover_95" (percentage of the catchment covered by each land cover class in 1995) and "cover_98" (percentage of the catchment covered by each land cover class in 1998).*

[Figure]

*Figure 3 - (a) 1995 land cover map and (b) 1998 land cover map prodived by the CNIGS.*

q. *Fig. 3, 5, 6, and 8 should be moved to supplement. This is not well explained and get in the way of the analysis.*

Figures 3, 5, 6 and 8 have been retained in the revised manuscript as they demonstrate that the use of relevant rain gauges has improved the performance and stability of the rainfall-runoff models used to simulate the streamflow series. These figures are essential for understanding the database.

r. *Fig 4 and 9 – The color pallet is not appropriate.*

An appropriate color pallet has been in the revised manuscript.

[Figure]

*Figure 7 - Spatial distribution of the average of the two KGE values obtained with GR2M in evaluation for the two subperiods. KGE values are calculated using (a) 20CR rainfall data, (b) reference raingauges (all raingauges) and (c) relevant raingauges. Dots represent catchments where model performance was improved by using the relevant raingauge combinations, triangles represent catchments where model performance was not improved by using the relevant raingauge combinations.*

[Figure]

*Figure 12 – Spatial distribution of observed mean annual streamflows (a) and simulated streamflows (b) with the GR2M parameters calculated over the entire period of available data.*

s. *Fig 10 – Why this figure? Is this an average for the entire country? How relevant for understanding the hydrology of the region is this?*

This figure shows rainfall and streamflow (observed and simulated) regimes. It shows i) the seasonality of rainfall and streamflow in Haiti, with two periods of heavy rainfall/streamflow around May and between September and November, and ii) a comparison of observed and simulated streamflow regimes. Each of the three regimes is calculated using data averaged over the entire country.

The figure has been revised. In addition to the averages over the whole country, ranges of values formed by the 5% and 95% quantiles have also been provided.

*Figure 13 shows the rainfall and streamflow regimes for the studied catchments. The results show a bi-modal rainfall/streamflow regime with two seasons of heavy rainfall/streamflow: the first season occurring around May and the second season between September and November, which corresponds to the cyclonic season. Rainfall is highly variable during the cyclonic season, with relatively heavy rainfall recorded in some catchments. The simulated streamflow represents well the seasonality of the observed streamflow (see Figure 13). However, simulated streamflows overestimate the observed values in May and underestimate them in November. In addition, the simulated streamflows slightly overestimate the low values in January.*

[Figure]

*Figure 13 – Seasonality of rainfall (obtained by combining the relevant raingauges) in blue, observed streamflow in black, and simulated streamflow with the parameters calculated over the entire period of available data in red. The ribbons Values ranges have been estimated using represent the range of values between the 10th and 90th percentiles, while the thick line represents the median values for the 24 catchments studied.*

t. *Fig 13 and 14 – Instead of this bars, isn't it better to provide a map of the geology and aquifer type of the region? Check most of the CAMELS papers for good examples.*

Bar charts have been replaced by maps and are presented in section 3.5.

The changes in the manuscript are:

*3.5.5 Geological attributes*

*The geological data provided by Butterlin (1960), Boisson and Pubellier (1987) and Terrier et al. (2014) have been used and have been made available by the CNIGS. The most common lithology types in Haiti are calcareous sedimentary rocks, followed by magmatic rocks (see Figure 4). The shapefile of lithology types has been cropped for each of the catchment studied (Table C4 shows the list of geological classes). The proportion of each lithology class in the catchment was calculated, corresponding to the "lithology" index. The proportion of carbonate rocks, sedimentary rocks and*

*magmatic rocks has been calculated for each of the catchment and corresponds to the "Carb_Rocks_Perc", "Sedim_Perc" and "Magma_Perc" indices.*

*3.5.6    Aquifer attribute*

*The aquifer data were produced by the MARNDR (Ministry of Agriculture, Natural Resources and Rural Development) in the 1990s have been used and have been made available by the CNIGS. Carbonate aquifers are the most widespread in Haiti, consist of carbonate rocks, mainly limestone and marl, and cover 53% of Haiti's surface area, of which karstic aquifers account for 18%. Crystalline formations, mainly magmatic rocks, account for 17%, alluvial aquifers for 16% and low-permeability sedimentary formations for 13%. Figure 4 shows the spatial distribution of the different aquifer classes and Table C4 shows the list of aquifer classes. The shapefile of aquifer classes has been cropped for each of the catchments studied. The proportion of each class in a catchment was then calculated, corresponding to the "aquifer" index.*

[Figure]

*Figure 4 - (a) lithological classes are represented by light colors for sedimentary rocks and shades of grey for magmatic rocks. (b) aquifer classes are represented by light colors for alluvial aquifers, blue colors for carbonate aquifers, and grey for crystalline aquifers.*

u.   *Fig 15 – This is a great example of useful graphics for having a quick look at the data and also precious information for decision makers.*

Thanks for this comment

**5.   CONCLUSION AND PERSPECTIVE**

a.   *L528 – "over several decades" seems to be a stretch when considering 1920-1940.*

"Over several decades" is replaced by "over several years".

b.   *L531 – Please, define and show in a figure what is meant by massive deforestation and anarchic urbanization, otherwise, delete it.*

"massive deforestation" and "the anarchic urbanization" has been deleted

c.   *L533-535 – Strange choice of words for these two sentences. Maybe delete?*

These two sentences have been reworded as follows:

*Frequency analysis methods can be utilized to estimate flood return periods. The accessibility of streamflow data allows for the possibility of various rainfall-runoff modeling approaches to be applied.*

*Haiti is acutely vulnerable to natural disasters, yet it lacks an accessible hydroclimatic database for analysis. The paper provides a meaningful database, known as Simbi, to fill this gap. The database is undoubtedly valuable, and the paper is well-written. I only have minor comments as follows:*

1. *it would be beneficial if the authors include a figure or a table detailing the data types, temporal extent, and other related information.*

   Table 2, which summarises the data used and produced in this study, has been added in section 3.5.

*Table 2 - Summary of the datasets used in this study.*

| Datasets | Source | Period of data availability |
|---|---|---|
| 156 monthly rainfall series | Moron et al., 2015 | 1905-2005 |
| 70 daily streamflow series. | BVH project | 1920-1940 |
| Paper archives contain daily rainfall series | BHS | 1920-1940 |
| Paper archives contain monthly air temperature series | BHS | 1920-1940 |
| NOAA 2OCR rainfall and air temperature daily database | Slivinski et al., 2019 | Twentieth Century |
| BEST air temperature database | Rohde et al., 2013 | Since 1753 |
| SRTM DEM with a resolution of 90 m | Reuter et al., 2007 | - |
| Shapefile of lithological classes on Haiti | CNIGS | - |
| Shapefile of aquifer classes on Haiti | CNIGS | - |
| Shapefile of land cover classes on Haiti | CNIGS | - |
| Shapefile of Haitian stream network | CNIGS | - |

*Table 3 – Summary of the datasets produced in this study.*

| Datasets | Period of data availability |
|---|---|
| Digitization of 59 daily rainfall series | 1920-1940 |
| Digitization of 23 monthly air temperature series | 1920-1940 |
| Rainfall, air temperature and PET series at catchment scale and at daily and monthly time steps for 24 catchments studied. | 1920-1940 |
| Simulated streamflow series at daily and monthly time steps for 24 catchment studied. | 1920-1940 |
| 49 attributes for each of the 24 catchment areas studied | - |

2. *In the Simbi database, some data have been sourced from previous studies. It is essential to explicitly describe the origins of the data within the database.*

   The origin of the hydro-meteorological data used in Simbi has been described in the introduction to the revised manuscript.

*Unfortunately, there are significant differences between countries in terms of the quality and quantity of hydroclimatic reference databases, as well as regarding access to these data. Some countries do not have such reference databases. This is the case of Haiti, whose territory is, moreover, highly exposed to natural disasters (Khouakhi et al., 2017; Burgess et al., 2018), and climate change (Peterson et al., 2002). At the same time, Haiti is facing the consequences of massive deforestation and anarchic urbanization (urban development that does not comply with planning regulations) in recent decades (Hedges et al., 2018; Tarter et al., 2018; Mompremier et al., 2022), resulting in increased vulnerability to hydroclimatic hazards. Currently, Haiti lacks a freely and easily accessible hydroclimatic database due to the absence of in situ hydroclimatic observations. The first hydrometric observations were conducted during the American occupation of Haiti, and began in 1919. American engineers from the Water Resources Service (WRS) of the United States Geological Survey (USGS) supervised these hydrological observations, that continued into the 1940s and exceptionally later. The end of the American occupation is the main reason for the cessation of hydrometric observations. This is due to the loss of technical support from the WRS, as well as financial constraints and socio-political difficulties in Haiti. The data time series and a description of the methods used to collect them were published annually in the "Hydrographic Bulletin", summarizing 70 daily streamflow time series over the 1920-1940 period. After these two decades of streamflow observations, very few hydrological data were produced in Haiti (Pouyaud and Hoepffner, 1987). In addition to hydrometric observations, rainfall measurements started in Haiti around 1905 using 15 raingauges. Over time, the raingauge network became denser, with 25 stations operated by the "Petit-Séminaire Collège St Martial" (a school run by the Congrégation du Saint-Esprit), 38 by the "Direction Générale des Travaux Publics", and nearly 30 by other institutions, such as the" Frères de l'Instruction Chrétienne" (Pouyaud and Hoepffner, 1987). Rainfall measurements are currently managed by the CNIGS and the UHM. Since 2014, this observation network has had approximately twenty automatic raingauges. However, due to a significant amount of missing data, the network remains highly fragmentary and unexploited.*

*In 1977, the Haitian government initiated a project to make an inventory and digitize some available hydroclimatic time series. As a result, the 70 daily streamflow series for the period 1920-1940 and almost a hundred monthly rainfall series from the start of observations (~1905) until 1975 were digitized. In 2012, the Haitian government launched a second project named BVH (Bassins Versants Haïtien in French, i.e., Haitian catchments; Gaucherel et al. 2018) for compiling available hydroclimatic data, better understanding hydrology in Haiti and improving the management of water resources. Within this project, Haitian catchments were characterized using monthly streamflow data (Gaucherel et al., 2016) and rainfall data (Moron et al., 2015) and the relationships between their shape, relief, and river sinuosity were investigated (Gaucherel et al., 2017, Bonhomme et al., 2013).Unfortunately, the two databases produced within the BVH project (monthly rainfall time series and monthly streamflow time series) have never been analyzed jointly, are not available online and remain limited for several hydrological analysis due to their monthly time step (monthly). Thus, these databases are underused to date.*

3. *In section 3.1.1, a rationale is needed to provide for selecting streamflow series that are hydrologically relevant.*

Several criteria were used to assess the quality of the flow series. These indices were used to select relevant streamflow time series.

Section 3.1.1 has been revised to better explain the flow series selection process.

*3.1.1        Selection of streamflow series*

*An analysis of the 70 available streamflow series was performed to select the "hydrologically relevant" streamflow series. Four criteria were initially used to make this selection:*

*1.  The annual hydrographic bulletins reported the accuracy with which rating curves were established through three ratings: "well established," "fairly well established," and "poorly established." Most of the streamflow series with "poorly established" rating curves were found to have significant measurement differences between periods. These streamflow series were not used in the remainder of this study.*

*2.  Some hydrometric stations were located downstream of diversion channels or small dams used for irrigation. These streamflow series poorly represent the seasonality of streamflow, and are therefore considered to be influenced by human activities. These streamflow series were not used in the remainder of this study.*

*3.  Some hydrometric stations were located downstream of resurgences or springs. These groundwater resurgences are beyond the scope of this study. Therefore, these streamflow series were not used in the remainder of this study.*

*4.  The streamflow series that had less than 5 years of data were not used in the remainder of this study.*

*In addition to these four criteria, three other indices inspired by the paper of Gudmundsson et al. (2018) were used to assess the quality of the streamflow data. These three criteria were calculated as follow:*

*1.  Number of days for which Q<0, where Q denotes a daily streamflow value. The rationale underlying this rule is that streamflow values smaller than zero are non-physical (Gudmundsson and Seneviratne, 2016).*

*2.  Sequence of more than 10 equal consecutive streamflow values larger than zero. This index was selected because equal consecutive streamflow values often occur due to instrument failure or flow regulation (Gudmundsson et al., 2018).*

*3.  Detection of outliers, i.e. unusually large or small streamflow values that could come from instrument malfunction. The calculation of these outliers is inspired by the papers of Gudmundsson et al. (2018): daily streamflow values are flagged as outliers if values of log (Q+0.01) are larger or smaller than the mean value of log (Q+0.01) plus or minus 6 times the standard deviation of log (Q+0.01) computed for that calendar day over the entire series. The mean and standard deviation are computed for a 5-day window centered on the calendar day to ensure that a sufficient amount of data is considered. The log-transformation is used to account for the skewness of the distribution of daily streamflow values and 0.01 was added because the logarithm of zero is undefined.*

*To summarize, the quality of the 70 streamflow daily series is described using 12 flags (1, 2, 3, 4, A, B, C, D, E, F, H and I), as detailed in the Table 1. Using these criteria, along with visual analysis to identify anomalies (i.e. non-natural records that may be erroneous streamflow values or anthropogenic influences that can lead to misinterpretation of actual hydrological processes (Strohmenger et al., 2023)), 24 hydrometric stations were identified as "hydrologically relevant" from the 70 available.*

4.  *In Section 3.2.1, the description of "multiple raingauge combinations" lacks clarity. If the number of rain gauges is fewer than three, it is unclear how to utilize Thiessen polygons.*

Section 3.2.1 has been revised to better explain the process of selecting the relevant rainfall series.

*3.2.1 Rainfall*

[revised manuscript text omitted]

6. *In Figure 10, the average monthly streamflows of 48 catchments were calculated. However, there is considerable uncertainty due to data discrepancies in many catchments. Therefore, it is recommended to include confidence intervals in Figure 10.*

Due to the variability of flows in the 24 catchment studies, a range of values representing the 5% and 95% quantiles have been added to the figure in addition to the median streamflows.

[Figure]

*Figure 13 – Seasonality of rainfall (obtained by combining the relevant raingauges) in blue, observed streamflow in black, and simulated streamflow with the parameters calculated over the entire period of available data in red. The ribbons Values ranges have been estimated using represent the range of values between the 10th and 90th percentiles, while the thick line represents the median values for the 24 catchments studied.*

7. *In Figure 10, providing reasons for the observed lags in the trend between rainfall and streamflow in Nov.*

Thanks for this comment, which could be the subject of future work on hydrology in Haiti. The second season of heavy rainfall in Haiti, from September to November, gradually moistens the soil until it is saturated. Although rainfall in November is relatively lower than in October, the steamflows in November are generally higher than in October due to soil saturation. However, this hypothesis needs to be further investigated in future work.

This analysis has been added to the revised manuscript.

*Figure 13 shows the rainfall and streamflow regimes for the studied catchments. The results show a bi-modal rainfall/streamflow regime with two seasons of heavy rainfall/streamflow: the first season occurring around May and the second season between September and November, which corresponds to the cyclonic season. Rainfall is highly variable during the cyclonic season, with relatively heavy rainfall recorded in some catchments. The simulated streamflow represents well the seasonality of the observed streamflow (see Figure 13). However, simulated streamflows overestimate the observed values in May and underestimate them in*

*November. In addition, the simulated streamflows slightly overestimate the low values in January. A time lag has been observed between the peak rainfall in October and the peak flow in November. This lag can be explained by soil saturation. The second season of heavy rainfall in Haiti, from September to November, gradually moistens the soil until it is saturated. Although the rainfall in November is relatively lower than in October, the streamflows in November are generally higher due to soil saturation. However, this hypothesis requires further investigation in future study.*

8. *According to Simbi, can you provide a summary of the hydroclimatic characteristics of this region?*

The hydrological characteristics of the catchments studied were summarised in the form of graphical summary sheets such as the one presented in section 4.6 for one of the catchments studied.

Nevertheless, this sentence has been added in section 4.5.1 in order to highlight the general hydrological characteristics of the region.

*Figure 13 shows the rainfall and streamflow regimes for the studied catchments. The results show a bi-modal rainfall/streamflow regime with two seasons of heavy rainfall/streamflow: the first season occurring around May and the second season between September and November, which corresponds to the cyclonic season. Rainfall is highly variable during the cyclonic season, with relatively heavy rainfall recorded in some catchments.*

---

## Author Response (AR2)

**Simbi: historical hydro-meteorological time series and signatures for 24 catchments in Haiti**

**Manuscript No. ESSD-2023-259**

Ralph Bathelemy, Pierre Brigode, Vazken Andréassian, Charles Perrin, Vincent Moron, Cédric Gaucherel, Emmanuel Tric, and Dominique Boisson.

**Answer to the reviewer 1**

Please find below our replies to the Reviewer's comments. We provided specific responses (in black) the reviewer comments (in *italic and blue*).

*The authors did a thorough job of responding to the reviewer comments and have done a lot of work to improve the manuscript. I suggest to publish the paper as is. The first folder level of the dataset should only contain folders except the readme file. Therefore, I suggest creating additional folders: e.g. 06_SIMBI simulation, 07_SIMBI_Codes, 08_SIMBI_metadata and add the corresponding csv files. I would rename the folder 05 to 05_SIMBI_WEIGHTS and add the file SIMBI_weights_temperature.csv. Folder 04 contains one underscore too much in its name "04__SIMBI_MAP".*

*Next time I recommend using Zenodo for data storage. There the folders are not compressed multiple times (tedious).*

We thank the Reviewer 1 for this detailed review and helpful comments. Your comments have been taken into account in the organisation of the database folders. The first folder level of the dataset currently contains 9 folders and the readme file. A folder 06_SIMBI_SIMULATION has been created containing the results of simulations with the GR2M and GR4J models. A folder 07_SIMBI_CODES has been created and contains the r-code file. A folder 08_SIMBI_METADATA has been created and contains the metadata of the rain gauges and flow stations. Folder 05_SIMBI_WEIGHTS_RAINGAUGE has been renamed to 05_SIMBI_WEIGHTS and the file SIMBI_temperature_weights.csv has been added to this folder. The extra underscore in folder 04_SIMBI_MAP has been removed.